# Light-driven self-assembly of spiropyran-functionalized covalent organic framework

Gobinda Das[1,11], Thirumurugan Prakasam[1,11], Nour Alkhatib[1], Rasha G. AbdulHalim[1], Falguni Chandra[2], Sudhir Kumar Sharma[3], Bikash Garai[1,4], Sabu Varghese[5], Matthew A. Addicoat [6], Florent Ravaux[7], Renu Pasricha[5], Ramesh Jagannathan [3], Na'il Saleh [2,8], Serdal Kirmizialtin[1,9], Mark A. Olson [10] ✉ & Ali Trabolsi [1,4] ✉

Controlling the number of molecular switches and their relative positioning within porous materials is critical to their functionality and properties. The proximity of many molecular switches to one another can hinder or completely suppress their response. Herein, a synthetic strategy involving mixed linkers is used to control the distribution of spiropyran-functionalized linkers in a covalent organic framework (COF). The COF contains a spiropyran in each pore which exhibits excellent reversible photoswitching behavior to its merocyanine form in the solid state in response to UV/Vis light. The spiro-COF possesses an urchin-shaped morphology and exhibits a morphological transition to 2D nanosheets and vesicles in solution upon UV light irradiation. The merocyanine-equipped COFs are extremely stable and possess a more ordered structure with enhanced photoluminescence. This approach to modulating structural isomerization in the solid state is used to develop inkless printing media, while the photomediated polarity change is used for water harvesting applications.

The self-assembly of organic nanostructured materials with tunable morphology has become a fascinating topic in material science and supramolecular chemistry[1–10]. However, supramolecular self-assemblies are formed by weak non-covalent interactions that are easily destroyed by heat or changes in pH, effectively limiting their practical utility[11–14]. To overcome these problems, scientists have introduced the bottom-up self-assembly process for the synthesis of 2D nanosheets[15], hollow spheres[16], and tubular morphologies[17] in covalent organic frameworks (COFs)[18]. These materials have attracted much attention due to their excellent chemical stability and strong mechanical properties[15]. Unfortunately, the preparation of COFs by the bottom-up self-assembly approach suffers from long reaction times and requires much effort to obtain product[15–17].

Triggering the self-assembly of materials to obtain a specific morphology with an external stimulus is another smart approach to control the self-assembly process[19–22]. This strategy is promising for those seeking to develop new well-defined nanoscale architectures in short time that also have potential applications in optics[23–26], sensing[27–29], and

[1]Chemistry Program, New York University Abu Dhabi (NYUAD), Saadiyat Island, Abu Dhabi, United Arab Emirates. [2]Chemistry Department, College of Science, United Arab Emirates University, P.O. Box 15551 Al-Ain, United Arab Emirates. [3]Engineering Division, New York University Abu Dhabi (NYUAD), Abu Dhabi, United Arab Emirates. [4]NYUAD Water Research Center, New York University Abu Dhabi (NYUAD), Saadiyat Island, Abu Dhabi, United Arab Emirates. [5]CTP, New York University Abu Dhabi, 129188 Abu Dhabi, United Arab Emirates. [6]School of Science and Technology, Nottingham Trent University, Clifton Lane, NG11 8NS Nottingham, UK. [7]Quantum research center, Technology Innovation Institute, P.O. Box 9639 Abu Dhabi, United Arab Emirates. [8]Zayed Center for Health Sciences, United Arab Emirates University, P.O. Box 15551 Al Ain, United Arab Emirates. [9]Center for Smart Engineering Materials, New York University Abu Dhabi (NYUAD), Abu Dhabi, United Arab Emirates. [10]Department of Physical and Environmental Sciences, Texas A&M University Corpus Christi, 6300 Ocean Dr., Corpus Christi, TX 78412, USA. [11]These authors contributed equally: Gobinda Das, Thirumurugan Prakasam. ✉e-mail: mark.olson@tamucc.edu; ali.trabolsi@nyu.edu

delivery systems[30–32]. Stimuli-responsive self-assembly of one- and two-dimensional supramolecular materials, including polyrotaxanes[33–40], nanotubes[41,42], nanowires[43–48], and nanosheets[43,49–52], has already been extensively studied[46,47,52,53]. However, the modulation of morphology in response to light has rarely been reported despite being one of the most environmentally friendly sources of external stimuli that is commonly accessible in our daily life[19,20,54,55]. Compared to other stimuli, light has numerous advantages because it can be applied at a specific location and in the form of different wavelengths[56,57]. An attractive strategy to control morphological self-assembly in response to light is to incorporate photoactive molecular switches that undergo structural isomerization upon light irradiation into the COF network[58–60]. Among the numerous photoactive molecular switches described in the literature, spiropyran (SP)[61] is one of the best studied molecular switches that can reversibly photoisomerize between the hydrophobic ring-closed SP and the hydrophilic ring-opened merocyanine (MC) upon irradiation with visible or UV light[56,62,63]. Due to their excellent reversible reactivity, materials functionalized with spiropyran have been widely used for data storage[64,65], chemical sensors[66–69] and developing smart materials[62,70,71]. However, on account of the space confinement effect and/or their structural rigidity, it remains a challenge to achieve reversible photoisomerization in ordered structures such as metal-organic frameworks (MOFs)[72–78] and COFs[58,59,79–81]. Recently, we developed a photosensitive quasi-honeycomb COF by introducing dangling azobenzene units into the pore structure[58]. The material exhibited interesting reversible photoswitching behavior due to the free movement of the azobenzene groups within the pores upon irradiation. Inspired by our previous work, we first attempted to intrinsically introduce spiropyran into a COF structure as a pendant group by following our reported synthetic protocol with the same 1:1.5 monomer:linker ratio. Interestingly, unlike the azobenzene COF, the spiropyran COF did not exhibit any photoswitching properties, which could be explained by the larger size of the spiropyran moiety as compared to the azobenzene unit, which hinders mechanical motion within the pores. To avoid the crowding effect, we used a mixed linker synthesis method (Fig. 1a) to reduce the number of spiropyran units and to facilitate the reversible photoswitching behavior within the pores. The mixed linker COF (TTA-SPDFP) exhibited a reversible photoswitching behavior (TTA-SPDFP ↔ TTA-MCDFP, Fig. 1a) in the solid state, in response to UV/Vis light exposure over several cycles. To date, only a limited number of porous materials have been reported with photoswitching properties in solid materials due to their close-packed structural arrangement. In this work, we also studied the morphological changes observed for TTA-SPDFP COF when exposed to UV light. The dispersed phase of TTA-SPDFP COF exhibited photo-controlled self-assembly from a rigid, spherical, urchin-like morphology to very small, hollow-structured vesicles that develop through the formation of 2D nanosheets as UV light-induced intermediates. The ability to tune the morphology of this material when exposed to UV light is attributed to the dramatic change in polarity and structure that can be achieved by photoisomerization of the spiropyran units within the COF. However, the change in morphology in response to UV light is an irreversible process that could be due to the stabilization of the MC form in the polar medium. The stabilization of the metastable MC form is important for many applications, as it provides an accessible docking site that could potentially serve as a potential sensor for toxic guest molecules[82–84], and as a tunable surface with varying polarity and wettability[85,86]. Crystalline COFs with different morphologies such as hollow spheres[16,87,88], tubes[17], and 2D nanosheets[89] have also been widely reported[18]. However, nanoscale vesicular morphologies have not yet been observed for COFs.

Overall, the controlled incorporation of photoresponsive molecular switches into COF materials introduces a new opportunity for the enrichment of reversible solid-state properties. Their incorporation into COFs facilitates the one-step, photoregulated self-assembly process that transforms rigid spheres into highly stable vesicles within an hour of UV light exposure. This work demonstrates how the carefully designed spiropyran-functionalized COF can rapidly undergo self-assembly when exposed to light, ultimately resulting in morphological changes that evolve over time. Additionally, the reversible and distinct color change associated with the solid-state structural isomerization has been utilized for developing facile media for inkless printing, a promising and green solution for temporary printing. Furthermore, the polarity change of the surface from the conversion of the SP to MC form shows another promising application of the COF as water harvester.

## Results

### Synthesis and structural investigations of TTA-SPDFP/MCDFP COF

The spiropyran-functionalized COF (TTA-SPDFP) was prepared via imine condensation of the monomers 4,4′,4″-(1,3,5-triazine-2,4,6-triyl)-trianiline (TTA), pyridine-2,6-dicarbaldehyde (DFP), and (R)-4-(1′,3′,3′-trimethylspiro[chromene-2,2′-indolin]-6-yl)-pyridine-2,6-dicarbaldehyde (SPDFP, Supplementary Fig. 1). The reactions were carried out using the sonochemical method in a 25 mL glass vessel at room temperature for 2 h, in the presence of acetic acid and using 1,4-dioxane as a solvent. The product was obtained as a yellow powder, which was subsequently purified by washing with ethanol, followed by drying at 120 °C for 12 h (Fig. 1a). The product was completely insoluble in water and common organic solvents. Sonochemical reactions were performed with a 550 W ultrasonic probe operating in continuous mode using a 3 mm microtip probe at 40% sonication power (the maximum power recommended for the used probe). The formation of TTA-SPDFP was confirmed by Fourier transform infrared (FT-IR) and $^{13}$C NMR spectroscopies.

As shown in the FT-IR spectra (Supplementary Fig. 2), the N-H stretching vibrations (3311 cm$^{-1}$ for TTA) and the C=O stretching vibrations (1700 cm$^{-1}$ for SPDFP and 1702 cm$^{-1}$ for DFP) disappeared while a new characteristic C=N stretching band at 1500 cm$^{-1}$ was detected, suggesting the formation of an imine bond. The $^{13}$C NMR spectra (CPMAS NMR, Supplementary Fig. 3) of TTA-SPDFP obtained in the solid state confirmed the formation of imine bonds, as evidenced by the resonance signal at 165 ppm. The signal at $\delta = 169.8$ ppm could be attributed to the triazine ring in the TTA-SPDFP COF. The subcomponent linker ratio of the TTA-SPDFP COF was confirmed by $^{1}$H NMR spectroscopy in solution following digestion of the COF in nitric acid at room temperature. The $^{1}$H NMR spectra of the acid-digested TTA-SPDFP COF was recorded in DMSO-$d_6$ at room temperature (Supplementary Fig. 4a, b, Supplementary Table 1), and the two-dimensional $^{1}$H-$^{1}$H COSY experiment (Supplementary Fig. 5) allowed us to fully assign all peaks in the corresponding $^{1}$H NMR spectrum of TTA-SPDFP COF.

The aromatic pyridine protons (b and g) of the acid-digested COF showed two distinct singlet signals at $\delta = 8.41$ and 8.19 ppm with an integration ratio of 2:1, corresponding to the DFP and SPDFP subunits, respectively. The TTA protons (e and d) of the acid-digested TTA-SPDFP COF resonance signals appeared as two separate doublet signals at $\delta = 8.61$ and 7.54 ppm, and the integration region of the TTA resonances showed a 2:2:1 ratio of TTA, DFP, and SPDFP compared with the precursor linkers. The acid-digested TTA-SPDFP COF also showed three characteristic aldehyde proton signals (c and f) at $\delta = 10.01$, 10.10, and 9.95 ppm, and all protons had the same integration area (2:0.5:0.5), which corresponded to the aldehyde protons of DFP and SPDFP. Due to the ring opening of the SPDFP subunit in the strongly acidic medium, the symmetry of SPDFP was lost, so that the two aldehyde protons appeared as two separate singlets at $\delta = 10.10$ and 9.95 ppm. Area integration of the remaining other protons (h-r) of the acid-digested TTA-SPDFP COF, corresponded to SPDFP, and showed good agreement with proton (a) of the DFP subunit. Together, these results support the formation of TTA-SPDFP with a linker ratio of

 

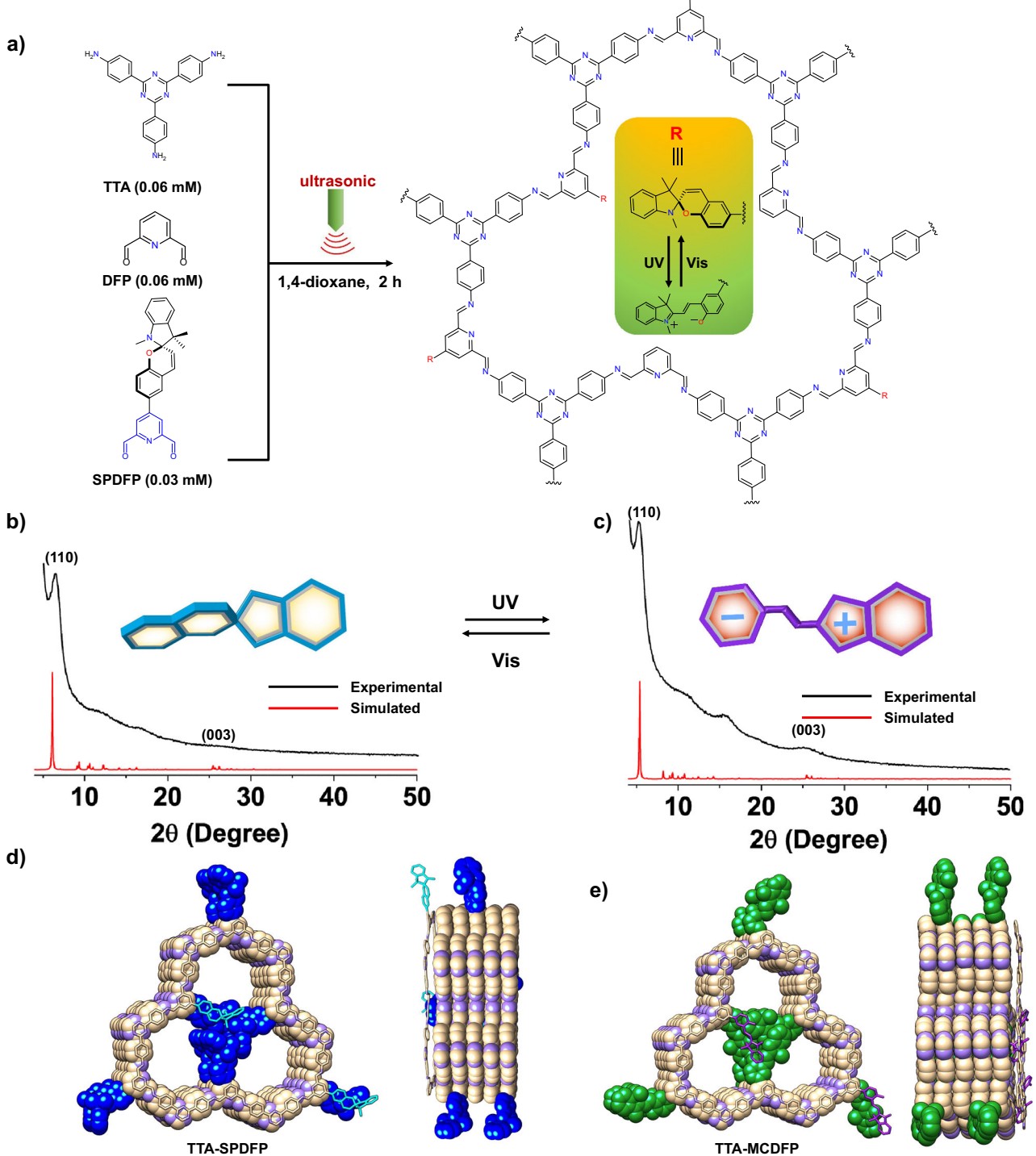

**Fig. 1 | Design and synthesis of spiropyran-functionalized COF. a** Synthetic scheme for obtaining TTA-SPDFP COF under ultrasonic waves at room temperature in 3 mL of 1,4-dioxane. **b, c** Powder X-ray diffraction (PXRD) patterns of the TTA-SPDFP COF and TTA-MCDFP COF. The insets show the cartoon images of the spiropyran and merocyanine. **d, e** Simulated crystal structures that resulted from an eclipsed AA stacking model of TTA-SPDFP and TTA-MCDFP COFs. For proton assignment of the NMR spectrum, see Supplementary Fig. 3a.

2:2:1 for the TTA, DFP and SPDFP subcomponents. Thermal stability of the as-synthesized COF was confirmed by thermogravimetric analysis (TGA), which revealed that the TTA-SPDFP is thermally stable up to $400 \pm 5\,°C$ (Supplementary Fig. 6) without any significant loss in weight.

Powder X-ray diffraction (PXRD) analyses were performed on the TTA-SPDFP COF (Fig. 1b) and its UV-irradiated open form, TTA-MCDFP COF (Fig. 1c). The XRD patterns of both COF samples showed similar peak positions of the 110 (2θ = 5.8°) and 003 (2θ = 26.1°) facets, indicating that the structure order and connectivity was preserved. The structural models were based on a trigonal quasi-honeycomb (hcb) layered topology with a $P$1 space group[58]. The experimentally obtained PXRD patterns and the calculated patterns for the COFs agrees well with the slipped eclipsed model, and the Density Functional Tight Binding (DFTB) optimized unit cell parameters were found to be a = 33.0 Å, b = 33.0 Å, and c = 10.4 Å. The proposed crystal structure of

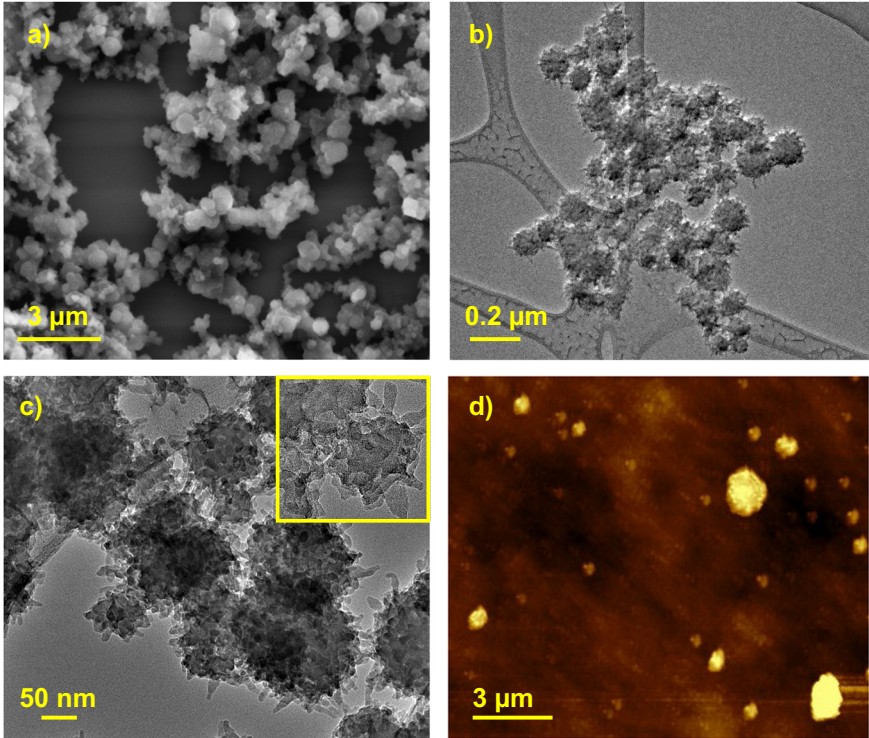

**Fig. 2 | Microscopic characterization of TTA-SPDFP COF. a** FESEM, **b**, **c** HRTEM, and **d** AFM micrographs of the TTA-SPDFP COF.

TTA-SPDFP (Supplementary data 1) indicates that the spiro group is twisted into an upright position when the molecule is in its closed form, as shown in Fig. 1d. This causes inefficient stacking of the 2D layers and leads to a distorted structure. The reduced crystallinity of the obtained COF is reflected in the weaker intensity of its PXRD peaks when compared to the light-irradiated merocyanine COF sample, TTA-MCDFP, which has a more planar structure (Supplementary data 2), as shown in Fig. 1e. The low intensity PXRD pattern of TTA-SPDFP COF can be attributed to the presence of the bulky non-planar spiropyran and methyl groups inside the pore channels, which are not planner and thus inhibit efficient π-π stacking. To confirm this hypothesis, an isostructural triazine-COF structure, TTA-DFP, was synthesized using the same synthetic protocol[90]. As revealed from PXRD analysis (Supplementary Fig. 7), TTA-DFP COF shows high crystallinity compared to TTA-SPDFP COF, likely due to the enhanced π-π stacking.

## Morphology of TTA-SPDFP COF

The size and morphology of the TTA-SPDFP COF were examined by field emission scanning electron microscopy (FESEM, Fig. 2a, Supplementary Fig. 8a–c), high-resolution transmission electron microscopy (HRTEM, Fig. 2b, c, Supplementary Fig. 8d–i), and atomic force microscopy (AFM, Fig. 2d, Supplementary Fig. 9). FESEM and HRTEM imaging revealed that the TTA-SPDFP COF had an urchin-shaped morphology[91] with an average diameter of 170 nm (Fig. 2a–c). At higher magnification, HRTEM images show that the 2D sheets are stacked on top of each other and form an urchin-like spherical morphology (Fig. 2c). AFM analysis confirmed the formation of the urchin-shaped morphology (Fig. 2d, Supplementary Fig. 9a–d) in agreement with FESEM and HRTEM images.

## Photoisomerization in the solid state

The solid-state photoswitching behavior of the COF materials was investigated by diffuse-reflectance UV-vis (DR UV-vis) spectroscopy. As shown in the inset of Fig. 3a, the yellow powder of TTA-SPDFP turned deep green within 5 min of continuous exposure to UV light (365 nm UV lamp, 1500 mA), an indication that photoisomerization of SP to MC

had taken place. The photoisomerization reaction of SP to MC promotes the expansion of the unit's π-conjugation, resulting in the appearance of a prominent absorption band in the visible region with $\lambda_{max}$ of 639 nm, indicating the formation of the merocyanine isomer[92]. The MC form of the isomer, TTA-MCDFP, reverts to its SP form, TTA-SPDFP, after 8 h of irradiation with visible light at 450 nm. The formation of merocyanine introduces strong electrostatic interactions between the layers resulting in slowing down the reverse photoswitching process[61]. The reversible photoirradiation experiments were repeated five times (Supplementary Fig. 10) without showing any loss in the material's photostability. The detailed photoisomerization process was further investigated by FT-IR spectroscopy (Fig. 3b). TTA-SPDFP shows a characteristic peak of the C-O$_{spiro}$ stretching band at -1290 cm$^{-1}$. Following 20 min of photoirradiation, the intensity of this band decreased significantly, providing further evidence for the formation of the merocyanine product. The formed merocyanine content was determined to be 48% by integrating the FT-IR signals[93]. To further elucidate the structural changes upon light irradiation, we performed $^{13}C$ CPMAS NMR spectral analysis on both isomers (Fig. 3c). Surprisingly, the one-dimensional $^{13}C$ CPMAS NMR spectra (Fig. 3c) of the TTA-SPDFP and TTA-MCDFP states were almost identical. Both spectra show a chemical shift around 193 ppm, which is characteristic of the N$^{\oplus}$=C bonded C as previously reported in similar solution-based NMR studies[94,95]. However, the intensity of this band is lower in the closed form as compared to the open isomer.

The presence of a signal with a chemical shift of 193 ppm in the closed state suggested that some of the spiro moieties may have opened up when exposed to humidity[96], hence the two forms of spiropyran always coexist. This is a very well-known phenomenon in materials associated with spiropyran[97,98]. The spiropyran moiety is very sensitive to moisture, and some of the spiro moieties can be converted to the merocyanine form in the presence of moisture in the absence of UV light. The two-dimensional $^{1}H$-$^{13}C$ heteronuclear correlation spectra (Fig. 3d) of TTA-SPDFP (black) and TTA-MCDFP (red) were also nearly identical and showed $^{1}H$-$^{13}C$ correlation signals between neighboring protons and carbons. However, the spectrum of TTA-MCDFP in the

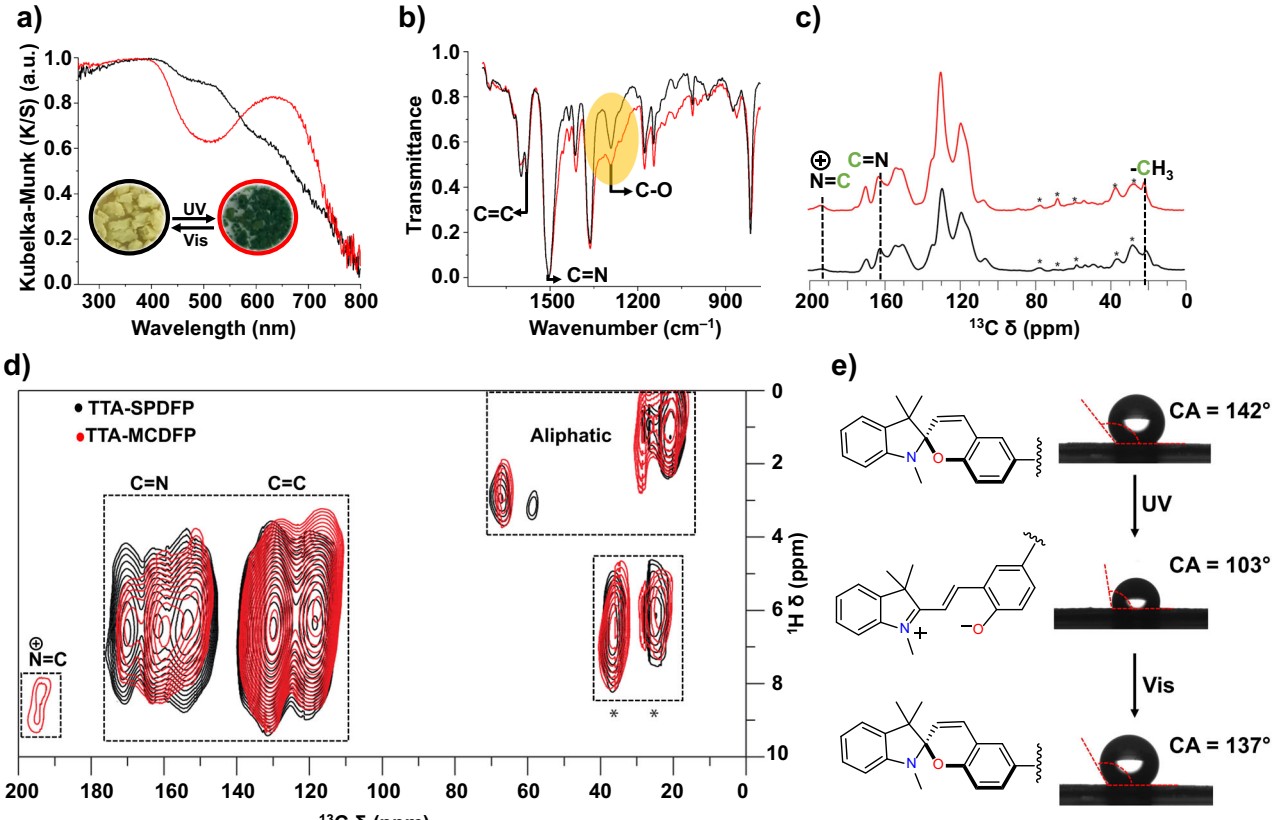

**Fig. 3 | Solid state photoswitching behavior of the TTA-SPDFP COF. a** Overlaid solid-state DR UV-vis spectra illustrating the light-induced spectral changes of the COF before (black trace) and after (red trace) exposure to 365 nm UV light irradiation, inset images of TTA-SPDFP COF showing the reversible color change upon UV-light irradiation. **b** Overlaid FT-IR spectra of the as-synthesized TTA-SPDFP COF (black trace) and the TTA-MCDFP COF (red trace) following UV light irradiation. The significant change in the C-O$_{spiro}$ vibration band upon UV irradiation is shaded in yellow. **c** Overlaid one-dimensional $^{13}$C CPMAS NMR spectra depicting the spectral changes of TTA-SPDFP COF before (black) and after photoisomerization (red). **d** Overlay of the two-dimensional $^{1}$H-$^{13}$C heteronuclear correlation (CPMAS NMR) spectra of TTA-MCDFP COF (red) and TTA-SPDFP COF (black). Both spectra were obtained using a MAS frequency of 14 kHz at 600 MHz ($^{1}$H Larmor frequency). Spinning side bands are indicated by asterisks. **e** Changes in the surface contact angle of TTA-SPDFP COF before and after UV light irradiation for 5 min followed by visible-light (450 nm) irradiation for 8 h.

open state (red) was characterized by the presence of a $^{1}$H-$^{13}$C correlation signal between 8 ppm ($^{1}$H) and 193 ppm ($^{13}$C), which is characteristic of the N$^{\oplus}$=C bonded C, as previously reported from similar solution-state NMR studies[94,95]. This signal was absent in the spectrum obtained for the closed state.

TTA-SPDFP COF exhibits reversible photoinduced surface wettability (Fig. 3e). Water contact angle measurements were performed on a TTA-SPDFP COF coated surface to verify its wettability behavior. The TTA-SPDFP COF is hydrophobic by nature and showed a contact angle of 142 ± 1° prior to light irradiation. The contact angle decreased to 103 ± 2° upon irradiation with 365 nm UV light for 5 min (Fig. 3e). The decrease in contact angle in response to UV light indicated the conversion of the nonpolar hydrophobic SP state to the hydrophilic and polar zwitterionic MC state. Reversibility was verified when the contact angle increased to 137 ± 2° when the sample was irradiated with visible light for 8 h. These results show that light can also control the wettability of the COF-coated surfaces functionalized with spiropyran.

The permanent porosity of TTA-SPDFP and TTA-MCDFP was determined by measuring N$_2$ adsorption isotherms at 77 K, as shown in Supplementary Fig. 11a, b. The surface area of the open form, TTA-MCDFP, displays a restricted adsorption behavior with Brunauer-Emmett-Teller (BET) surface area of 92 m$^2$g$^{-1}$ compared to the closed form, TTA-SPDFP, with a surface area of 334 m$^2$g$^{-1}$. As additional evidence of the conversion of SPDFP to MCDFP, non-local density

functional theory calculations (NLDFT) show a reduction in the total pore volume from 0.28 cm$^3$g$^{-1}$ to 0.13 cm$^3$g$^{-1}$, respectively, though no drastic changes were observed in the average pore sizes, as shown in Supplementary Fig. 11c, d. The decrease in the BET surface area in the merocyanine (MC) form is expected due to the change in the size of the molecule as well as the orientation from orthogonal for SP to planar for MC which is more exposed into the pores resulting in partially blocking the pore channels as shown in Supplementary Fig. 12a, b.

### Solution phase light induced self-assembly

In order to study the morphological changes of the synthesized COFs before and after irradiation with UV light, SEM, TEM, and AFM were performed on TTA-SPDFP and TTA-MCDFP COF samples dispersed in aqueous solutions. Microscopy analysis clearly shows that the particles of TTA-SPDFP COF undergo a distinct morphological change (Fig. 4a–d, Supplementary Fig. 13a–f). The spherical, urchin-shaped TTA-SPDFP COF particles (Fig. 4a, b) self-assembled into vesicle-like structures following UV light irradiation (Fig. 4c, d). SEM analysis revealed the formation of spherical vesicles after one hour of UV light irradiation, which is evident from the visible hollow, fractured inner cavity (Fig. 4c, Supplementary Fig. 13d–f). The formation of nanovesicles was also evident in HRTEM images, which showed the presence of spherical particles with dark exteriors and light interiors (Fig. 4d, Supplementary Fig. 14a–f), which are characteristic microscopy

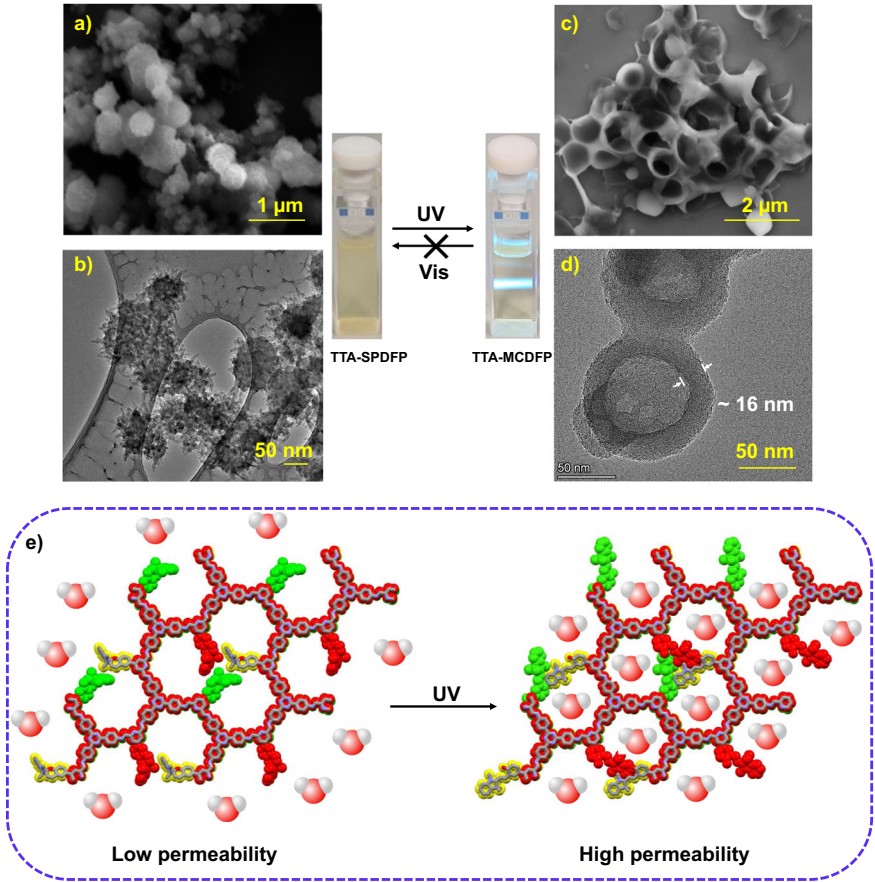

**Fig. 4 | UV-mediated change in morphology and structure of TTA-SPDFP COF.** SEM and HRTEM images of the TTA-SPDFP COF before (**a**, **b**) and after (**c**, **d**) UV light irradiation for two hours. In the middle, digital photographs show the aggregated (left) and homogeneous dispersion (right) of the TTA-SPDFP/MCDFP COF samples before and after photoirradiation. **e** Scheme of the effect of UV light on water permeability in TTA-SPDFP COF and TTA-MCDFP COF.

features of hollow vesicles[99–102]. AFM analysis was also performed on dried samples in an effort to understand the vesicle formation (Supplementary Fig. 15).

AFM imaging showed the presence of flattened cup-shaped particles with a height-to-diameter ratio of up to 1:33, corresponding to a thin-layered vesicle structure with a central collapsed region (Supplementary Fig. 15a–i)[101]. The morphological transformation was found to be irreversible, as vesicle formation under UV light led to the formation of the highly hydrophilic merocyanine form. Conversion to the merocyanine form facilitated the diffusion of the hydrating medium through the COF network (Fig. 4e) while simultaneously rendering the merocyanine state extremely stable in water due to its strong electrostatic attraction to water molecules[103]. Irreversible self-assembly in response to light is also known in the literature[19,104]. Upon continuous photoirradiation, solutions of the TTA-SPDFP COF particles gradually lose their turbidity becoming more transparent indicating a structural transformation to a much smaller particle size (Fig. 4 middle). The size of these vesicles is ~100 nm, much smaller than the as-synthesized urchin-shaped morphologies which had a particle size of ~170 nm. Furthermore, TEM analysis revealed a thickness of ~16 nm for the dark regions as shown in Fig. 4d. According to molecular simulation, the thickness of a single layer sheet is 0.5 nm, which suggested that the vesicle walls were comprised of multi (~32) stacked layers of COF network (Supplementary Fig. 16a–c). Furthermore, a well-defined Tyndall effect was observed for solutions of TTA-MCDFP COF-based vesicles under laser irradiation (Fig. 4 middle), an indication that the irradiated sample was comprised of well-dispersed homogeneous small-sized vesicular particles. On account of the closed form of the photoactive unit having a nearly perpendicularly bent conformation,

and its photoinduced open-ring isomer having a perfectly planar structure, such a large structural change at the molecular level could induce the observed dramatic morphology variations following irradiation. We propose a mechanism for this morphological transformation in response to light on the basis of the postulate that the TTA-MCDFP COF has a hydrophobic outer surface and a hydrophilic inner surface directed towards the pore. This difference in hydrophobicity of the interior and exterior surface of the TTA-MCDFP COF plays a major role in vesicle formation. Moreover, owing to the porous structure and hydrophilic nature of the TTA-MCDFP COF, water molecules can readily diffuse into the COF network and trigger the morphological transition.

In order to further investigate the mechanism for vesicle formation, we monitored the morphological changes (Fig. 5a) in response to UV light at different time intervals (Fig. 5b–j). After 10 min of irradiation with a 365 nm UV light, the urchin-shaped TTA-SPDFP COF (Fig. 5b, e, h) transformed into thin 2D sheet-like structures (Fig. 5c, f, i, Supplementary Fig. 17a–f). The nanosheets were very flat and the height of the sheets was about 5 nm (inset: Supplementary Fig. 16f). Following continuous photoirradiation for one hour, the turbidity of the dispersed COF solution became more evident, and all microscopic analyses were performed on the same solution. After prolonged photoirradiation of one hour, SEM, HRTEM, and AFM imaging revealed that the COF morphology changed, forming vesicles, as evidenced by the presence of hollow vesicle structures (Fig. 5d, g, j). A similar type of hollow vesicle morphology has already been observed for MOFs and other supramolecular polymers[99–102]. After an irradiation time of 24 h, further morphological changes were not observed (Supplementary Fig. 18a, b). However, the isolated COF vesicles were found to be stable

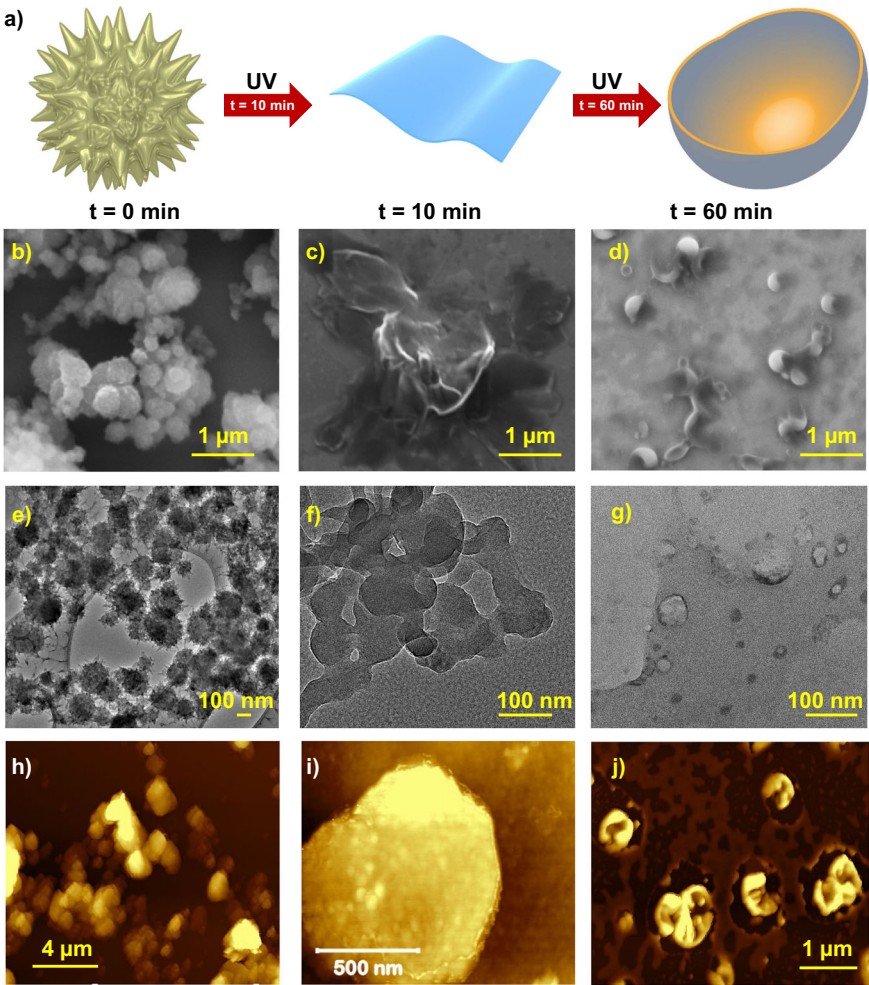

**Fig. 5 | Monitoring the morphological changes during the vesicle formation.** **a** Graphical representation illustrating the time-dependent vesicle formation stemming from morphology changes from urchin-shaped particles to 2D sheet-like structures to vesicles. Time-dependent (**b**–**d**) SEM, (**e**–**g**) HRTEM, and (**h**–**j**) AFM images following UV light irradiation at time = 0 (**b**, **e**, **h**), time = 10 min (**c**, **f**, **i**), and time = 60 min (**d**, **g**, **j**).

for several months, indicating their extreme stability in polar medium (Supplementary Fig. 19a, b).

Together, all these observations clearly confirm that the formation of hollow, structured vesicles is achieved by light-induced self-assembly in solution and not by the bottom-up mechanism of Ostwald ripening. Instead, we propose the following hypothesis for vesicle formation: As the UV irradiation time increases, the urchin-shaped morphology of TTA-SPDFP COF, formed by stacked 2D sheets, degrades in aqueous solution and can move freely in solution. These 2D sheets eventually coil into vesicle-like nanoassemblies driven by the electrostatic interactions between the polar charges of their merocyanine units. The self-assembly of 2D nanosheets into hollow spheres has also been reported in the literature for COFs formed by a boronic acid condensation reaction via the self-curling process[105].

However, the self-assembly process in our COF system can be precisely controlled by UV light. As a control experiment, we also tested the self-assembly process with other light sources of different wavelengths, but could not detect any morphological changes (Supplementary Fig. 20b). To verify the role of the aqueous solvent, we studied the morphological changes in the nonpolar solvent hexane, but could not detect any noticeable morphological changes in response to UV light exposure (Supplementary Fig. 20c). Therefore, both the light source and the polarity of the solvent are crucial factors in triggering the formation of stable vesicles.

To understand the molecular mechanism behind vesicle formation, we conducted a series of all-atom molecular dynamic simulations. We constructed six-layer COF structures using the TTA-SPDFP and TTA-MCDFP initial structures, with box dimensions extending to $19.6 \times 16.8 \times 9.5$ nm³. Figure 6a shows our explicit water simulation setup. The relatively large systems are aimed at observing macroscopic changes reported experimentally. We kept the pressure and temperature at 1 atm and 298 K, respectively, while allowing the simulation box dimensions to change. We monitored the structure of the COFs in a vacuum and in water. Details of our simulation setup are given in the supplementary materials. Figure 6b shows the initial and the equilibrium structure of TTA-SPDFP in a vacuum compared to TTA-MCDFP in water (Fig. 6c). Consistent with our experiments, we observed notable differences between the two COFs. Interestingly, TTA-SPDFP remained mostly planar with some structural undulations, while TTA-MCDFP in water showed notable bending (Fig. 6b versus 6c). To quantify the curvatures forming, we computed the average curvature from the trajectories. We observe higher curvature in TTA-MCDFP in water compared to TTA-SPDFP in a vacuum (Fig. 6d) further supports the experiments. Once we obtained good agreement between experiments and simulations, we examined the underlying mechanisms behind the formation of curvature in the case of TTA-MCDFP in water. To do so, we analyzed the changes in the solvent environment and the atomic structure of the COF layers. We compared TTA-SPDFP in a vacuum, TTA-SPDFP COF in water, and TTA-MCDFP COF in water

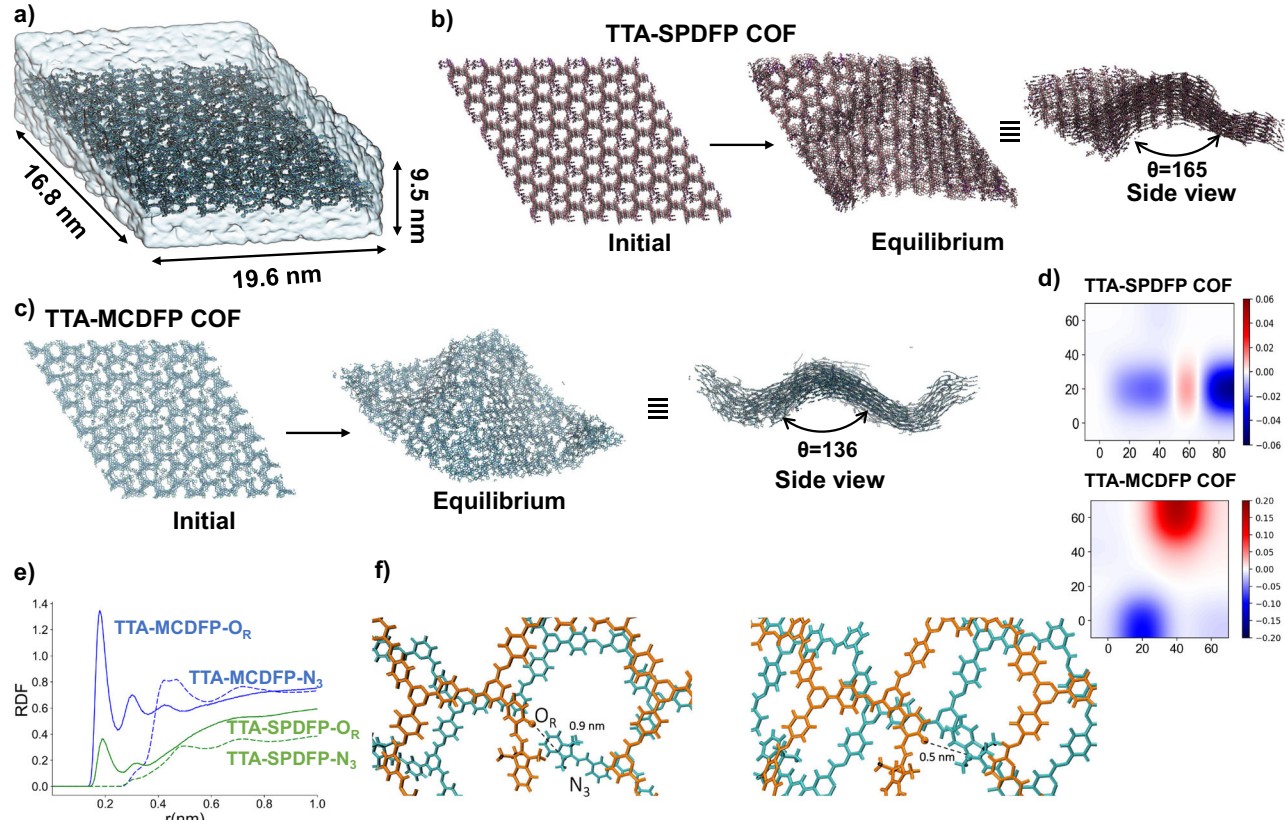

**Fig. 6 | Molecular dynamics simulation study of TTA-SPDFP and TTA-MCDFP in water and in vacuum. a** Simulation box of the molecular dynamics set up with TTA-MCDFP in water, **b** TTA-SPDFP in vacuum initial structure and its structure after equilibrium is reached (top and side views). **c** Same as B this time for TTA-MCDFP in water. **d** The structure of COFs is characterized by the average curvature plotted for the x-y plane. The color bars show the Gaussian curvature value for the two COFs, the larger the value the larger the curvature. **e** Radial distribution function of $O_R$ and $N_3$ atoms of the COF with water. **f** The conformational change of TTA-MCDFP layers leading to curvature formation in water observed during simulations.

to elucidate the roles of water and non-bonded interactions. Radial distribution functions computed between the atom types of the COFs and water showed that TTA-SPDFP is hydrophobic in nature (Supplementary Fig. 21). Interestingly, upon a change in the covalent structure (TTA-SPDFP to TTA-MCDFP COF), two sites became hydrophilic in nature (Supplementary Fig. 22 and Fig. 6e). The atoms in the linker ($O_R$ and $N_3$) switched from non-polar to polar (Fig. 6e), resulting in higher flexibility of the region. As a result of the increased polarity and opposite charge that the two atom types possess, we observed a shift in the stacked layers, such that to stabilize the layers the average distance between the two closest $O_R$-$N_3$ pairs changed from 0.9 nm to 0.5 nm (Fig. 6f). We found that this inter-layer electrostatic bond formation activated by photo-excitation slides the layers, resulting in an overall change in interlayer stacking leading to curvature formation in TTA-MCDFP COF.

### Crystallization-induced emission and acidochromic properties

To further investigate the effect of light irradiation on the emission properties of our material, powdered TTA-SPDFP COF was dispersed in ethanol and showed weak yellow emission centered at $\lambda_{max} = 525$ nm when excited with $\lambda_{ex} = 365$ nm (Fig. 7a, b). The weak emission of the closed form is the result of the orthogonal conformation of the photoemissive unit, leading to loose packing within the network. Upon continuous UV light irradiation for one hour, a significant increase (6 fold) in fluorescence emission was observed (Fig. 7a) along with a large blue shift of the emission maximum ($\lambda_{max} = 434$ nm). This can be attributed to the higher degree of conjugation and the planar structure of the merocyanine isomer in the TTA-MCDFP COF. To obtain further details on the correlation

between fluorescence enhancement and structural changes following light irradiation, HRTEM was recorded at a very high magnification to detect any profound structural transformation processes in response to UV light (Fig. 7c, d). Prior to irradiation, it was observed that the stacked nanosheets are disordered and irregularly arranged (Fig. 7c) due to the non-planar structure of the TTA-SPDFP COF. Following one hour of photoirradiation with a 365 nm UV light, a highly ordered TTA-MCDFP COF with clear hexagonal periodic channel-like features were observed by HRTEM, as shown in Fig. 7d. In addition, the ordered structure of the TTA-MCDFP COF was confirmed by selected area electron diffraction (SAED), which gave rise to numerous bright diffraction spots (Fig. 7e). In the ordered state, the intramolecular rotations and vibrations of the stacked COF layers are significantly constrained, thus enhancing the radiative transitions in the crystalline state that give rise to the improved emission property. This observation suggested that light plays an important role in controlling the enhanced crystallinity and emission properties of the material. Photoinduced crystallization with emission enhancement has not been previously reported for porous COFs.

The TTA-MCDFP COF also produced a reversible acidochromic response, whereby the phenoxide ion in its zwitterionic state undergoes protonation upon treatment with HCl gas, resulting in a deep brown-colored TTA-MCH⁺COF (Fig. 7f right). The protonated form of the COF was reversed by the very rapid passage of gaseous $NH_3$ which abstracts the protons from the TTA-MCH⁺DFP COF and regenerates the TTA-MCDFP COF (Fig. 7f left).

The acidochromic behavior was monitored experimentally by UV-vis spectroscopy. The UV-vis spectrum of an aqueous suspension of TTA-MCDFP COF showed the presence of an absorption band at

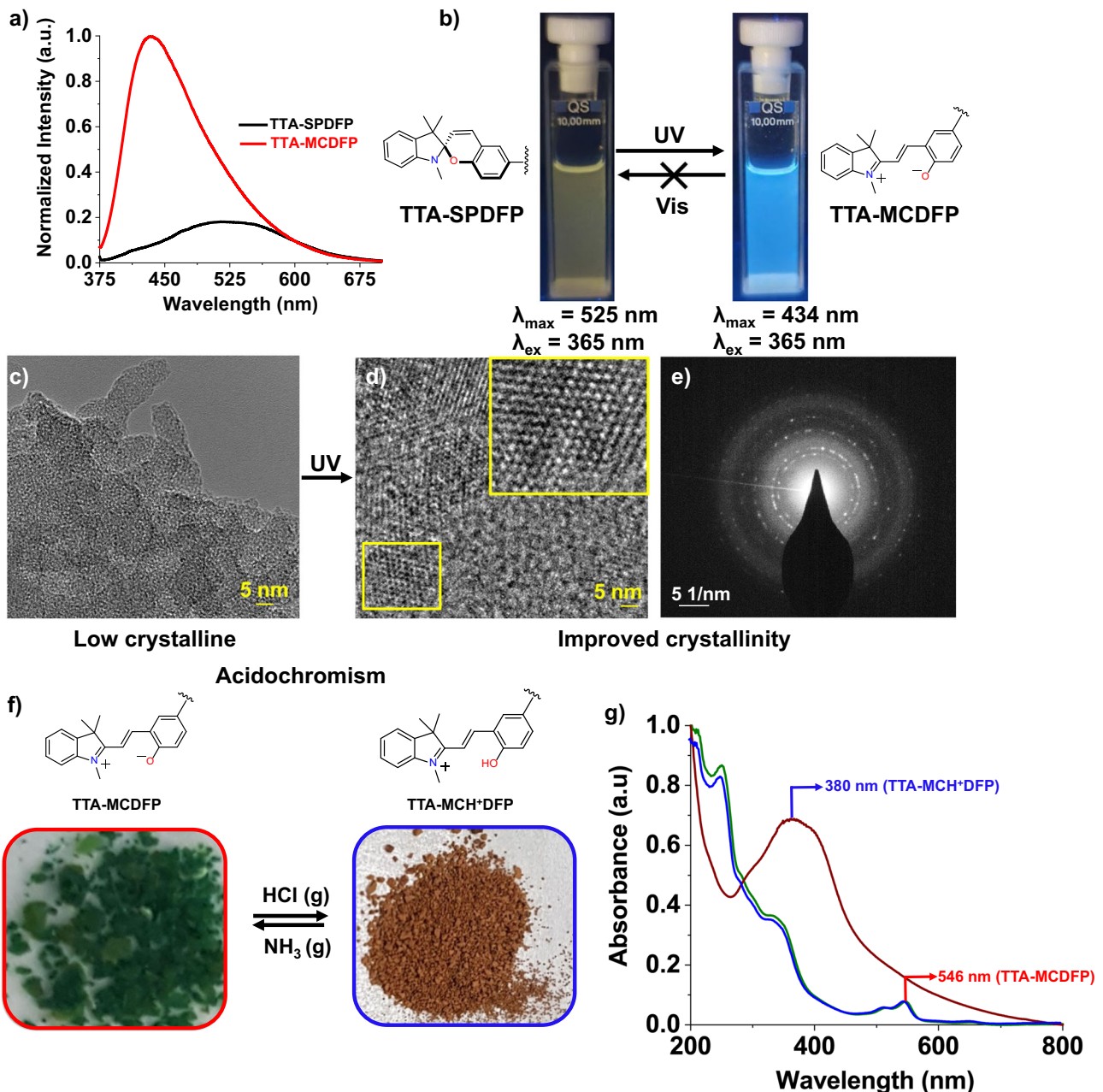

**Fig. 7 | Light-triggered photoluminescent and acidochromic properties of TTA-SPDFP/MCDFP. a** Emission spectra of TTA-SPDFP COF before (black) and after (red) photoirradiation, revealing significant emission intensity enhancement upon exposure to UV light for one hour. The spectra were recorded at room temperature with an excitation wavelength of 365 nm. **b** UV light induced visual emission color changes for TTA-SPDFP COF (left) and TTA-MCDFP COF (right) dispersions in ethanol. The process is not reversible with visible light. **c, d** HRTEM images of TTA-SPDFP/MCDFP COF and **e** SAED pattern of TTA-MCDFP COF. **f** Chemical structures (top) and photographs (bottom) depicting the reversible protonation and deprotonation of TTA-MCDFP COF powder in the presence of gaseous HCl and NH₃. **g** UV-vis absorbance spectral changes of TTA-MCDFP COF dispersed in water.

546 nm (Fig. 7g). After exposure to HCl vapor, the band at 546 nm disappeared and a new band appeared at 380 nm, corresponding to the formation of TTA-MCH⁺DFP COF, a behavior that has been reported for other spiropyran functionalized materials[106,107]. Reversible acidochromic switching of the material did not alter the structural integrity of the COF. The structural integrity was confirmed by the unperturbed SEM images and PXRD patterns of both the protonated and zwitterionic forms of the COF (Supplementary Fig. 23a–c). It is noteworthy to mention that this reversible color change phenomena upon exposure to external chemical stimuli makes this a very attractive system for use in visual chemical sensing applications.

## Light-induced inkless printing, and water harvesting applications

Due to the excellent switching reversibility of TTA-SPDFP ↔ TTA-MCDFP COF (Fig. 8a, b), the COF material is more suitable for temporary printing applications. Uniform dispersion of TTA-SPDFP COF was achieved by immersing the COF powder in ethanol under vigorous ultrasonication for 20 min. Due to the high dispersibility of the COF material, it can be uniformly coated onto the surface of a cellulose filter paper strip. The paper strip was air dried overnight at room temperature. Then, the coated paper was irradiated with a 365 nm UV light for a few seconds through a honey-bee designed stencil mask. The stencil was then removed from the surface of the coated paper,

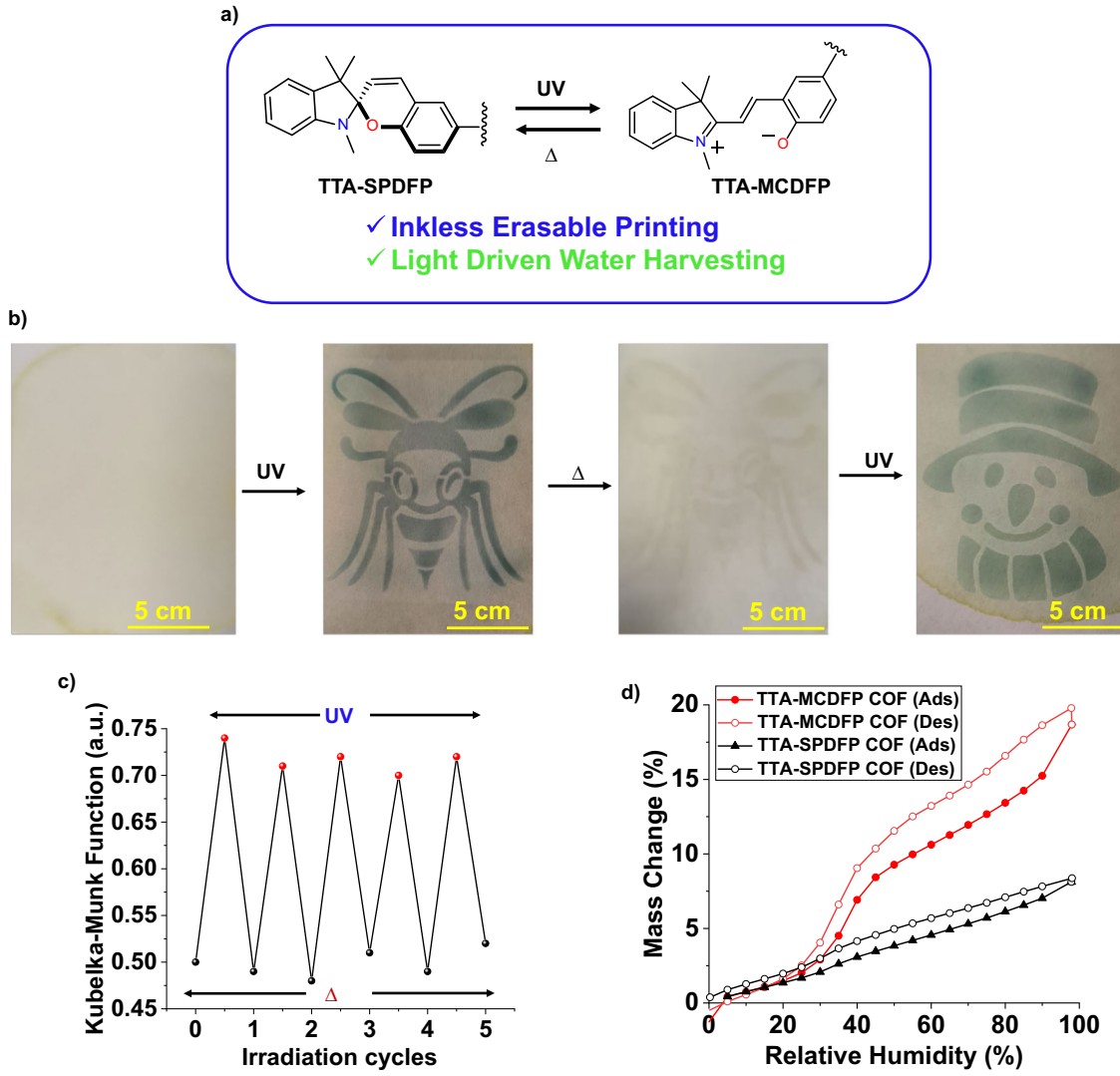

**Fig. 8 | Applications of TTA-SPDFP/MCDFP COFs. a** Schematic representation of the reversible, light/heat-promoted isomerization of TTA-SPDFP COF. **b** Photographs of Whatman 40 filter paper coated with TTA-SPDFP COF powder ($5 \times 5\,\text{cm}^2$) and the images were created sequentially in the same piece of paper by irradiating through different designed masks by alternating light/heat (365 nm light

for 5 s; $\Delta = 60\,°C$). **c** Reversible switching cycles in absorbance at $\lambda_{max} = 664$ nm of TTA-SPDFP COF in response to UV light (365 nm, 5 s) and thermal heating (60 °C, 60 s). **d** Water adsorption-desorption isotherms for the TTA-SPDFP and TTA-MCDFP COF's powder as a function of relative humidity.

revealing a clear, well resolved deep green-colored print of the insect on the COF-coated paper (Fig. 8b), which is due to the formation of the merocyanine form (TTA-MCDFP COF). Due to the exceptional photostability of the TTA-MCDFP COF's metastable form, the printed pattern can be kept for 2 days without fading of the color (Supplementary Fig. 24). Reversible isomerization of TTA-MCDFP → TTA-SPDFP could also be achieved by thermal annealing of the COF-coated test strip. Thus, the printed content can be quickly erased by heating the strip for 1 min (60 °C) if needed, which could be used to print another designed mask (Fig. 8b, c). Therefore, our photochromic COF, which acts as an erasable ink, can be effectively used as a convenient and easy to use solution for temporary printing of tattoos, advertising stickers and apparel designs.

Furthermore, we studied the water adsorption properties of the obtained COFs, since the hydrophobicity/hydrophilicity of TTA-SPDFP and TTA-MCDFP COFs affects the photoswitchable system. We performed water vapor adsorption analysis at room temperature on both the TTA-SPDFP and TTA-MCDFP COFs (Fig. 8d). The conversion of the highly hydrophobic form of TTA-SPDFP to the more hydrophilic form of TTA-MCDFP was illustrated by the change in the water adsorption

isotherm from a typical type VII[108] water adsorption isotherm with a total uptake of around 8%, as shown in Fig. 8d (black), to an S-shaped type V[108] isotherm (Fig. 8d, red) showing enhanced sorbent-water interaction with a total uptake of around 20% at 97% RH. These results confirm that the indoline cation in merocyanine is more hydrophilic compared to the as-synthesized material, which promotes water uptake in the COF network. The water adsorption studies were repeated three times to verify the stability of the TTA-MCDFP COF (Supplementary Fig. 25). To our knowledge, light-mediated water adsorption in COFs is not yet known and our light responsive TTA-MCDFP COF will open new avenues for the development of photo responsive water harvesters.

## Discussion

To summarize, we succeeded in synthesizing a porous, crystalline and photosensitive COF with defined spatial arrangement of the spiropyran units in the pores by a sonochemical method. The controlled photoisomerization was achieved by employing the mixed linker synthetic method to avoid the displacement effect of the molecular switch. Our strategic approach resulted in a spiro-COF that exhibited

exceptional reversible photoswitching properties in the solid state without any restriction. The TTA-SPDFP ↔ TTA-SPDFP photoreversibility was facilitated by switching a smaller number of spiropyran units within the pore channel, bypassing the potential for a molecular displacement effect. When dispersed in solution, the TTA-SPDFP COF resulted in unprecedented vesicle formation with enhanced luminescence properties and improved crystallinity upon light irradiation. The material was found to undergo a light-induced, time-dependent three-step COF self-assembly process that formed hollow vesicle structures with high stability that did not proceed via the bottom-up mechanism of Ostwald ripening. Finally, the open form of TTA-MCDFP COF showed excellent reversible acidochromic behavior without any structural degradation. This fast and reversible color change of the COF from solid state structural isomerization has been utilized for development of facile media for inkless printing. While the change in surface polarity from photoswitching of the SP to MC form has been exploited in water adsorption.

## Methods

### General material and instrumentation methods

All chemicals and reagents were purchased from Sigma-Aldrich and used without further purification. The 4-bromopyridine-2,6-dicarbaldehyde[58] (2c) and the pyridine-2,6-dicarbaldehyde (DFP)[109] were synthesized following previously reported synthetic procedures. Thin-layer chromatography (TLC) was performed on 60 $F_{254}$ silica gel TLC plates. The plates were inspected with UV light of wavelengths 256 nm and 302 nm. Column chromatography was performed using the Combi flash EZ Prep system. Nuclear magnetic resonance (NMR) spectra were recorded on a Bruker Avance III spectrometer, with working frequencies of 500 MHz for $^1$H, and 125 MHz for $^{13}$C nuclei at room temperature. All chemical shifts are given in ppm relative to the signals corresponding to the residual non-deuterated solvents ($CHCl_3 = 7.26$ ppm)[110,111]. All $^{13}$C spectra were recorded with the real-time decoupling of proton nuclei. The values of the coupling constants ($J$) are given in hertz (Hz). The multiplicity of the proton spectrum is abbreviated as follows: s (singlet), d (doublet), dd (doublet of doublets), t (triplet), and m (multiplet). High resolution mass spectrometry (HRMS) studies were done using an Agilent 6540 UHA Accurate Mass Q-TOF/LC-MS-spectrometer in the ESI positive ion mode using an acetonitrile/water a gradient with C18 column. All solid-state NMR experiments were carried out on a Bruker Avance-HD 600 MHz spectrometer operating at a static field of 14.1 T, resonating at 150.0 MHz for $^{13}$C, using a 3.2 mm triple resonance MAS probe. Fourier transform infrared (FT-IR) studies were carried out on the Agilent 670-IR spectrometer. Thermogravimetric analysis (TGA) was performed on TA SDT Q600. Scanning electron microscopy (SEM) images were obtained from FEI Quanta 450FEG. The topography of the TTA-SPDFP/MCDFP COFs were analyzed by dynamic atomic force microscopy (5500 Atomic Force Microscope; Keysight Technologies Inc., Santa Rosa, CA). We acquired topography, phase, and amplitude scans simultaneously. Silicon cantilevers (Nanosensors™, Neuchatel, Switzerland) with resonant frequencies of 250–300 kHz and force constants of 100–130 Nm$^{-1}$ were used. The set point value was maintained at 2.5 V. AFM scans were collected at 1024 points/lines with scan speed of 0.20 at fixed scan angle of 0º. Scan artifacts were minimized by acquiring a typical scan at an angle of 90º under identical image acquisition parameters. We used Gwyddion™ free software (version 2.47), SPM data visualization and analysis tool for post-processing the AFM scans. Size and morphology of the COF material was determined with a TEM (FEI-Titan 300) microscope. Samples were prepared on a carbon-coated copper grid. A drop of dispersed TTA-SPDFP/MCDFP COFs network was spotted on the grid and allowed to dry overnight. Surface area measurements were conducted on a Micromeritics 3Flex gas sorption analyzer. Samples (20–40 mg) were degassed at 85 °C for 24 h and then backfilled with $N_2$. Adsorption isotherms were generated by incremental exposure to ultrahigh-purity nitrogen up to 1 atm in a liquid nitrogen bath, and surface area was determined using BET adsorption models included in the instrument software (Micromeritics ASAP 2020 V4.00). Powder X-ray diffraction (PXRD) measurements were carried out using the PANalytical X'Pert PRO MP X-ray diffractometer consisting of a focusing elliptical mirror and a fast-high resolution detector (PIXCEL) with a radiation wavelength of 0.15418 nm. Solid state UV-vis studies were carried out on the Cary 5000 UV-vis-NIR spectrophotometer. Emission spectra in water at room temperature were recorded on a Perkin Elmer LS55 Fluorescence Spectrometer.

Water vapor adsorption experiments were carried out to examine water vapor adsorption characteristics of TTA-SPDFP /MCDFP COF using IGAsorp vapor sorption analyzer. The water vapor partial pressure was controlled automatically by mixing wet vapor feed with a dry $N_2$ line; thus, $N_2$ acts as a carrier gas for water vapor. Pre-drying of the sample was carried at 100 °C in the presence of $N_2$ dry carrier. The sample "dry mass" was measured under $N_2$ and was at equilibrium (25 °C) before dosing water vapor progressively into the chamber. The adsorption isotherms, obtained at equilibrium, were collected between 0 and 98 % RH at room temperature. The maximum equilibration time for each RH was maintained for 2 h.

### Computational method

Geometries of monolayer, AA, AB and slip-stacked TTA-SPDFP COF and TTA-MCDFP COFs with 33, 66 and 100% functionalized linkers were built from fragments using AuToGraFS[112] and optimized using Density Functional Tight Binding as implemented in DFTB+ version 19.1[113]. All atom pairs were described using standard parameters from the mio-0-1 parameter set[114]. Lattice parameters were optimized at the same time as atomic coordinates.

### General setup of molecular dynamic simulation

All-atom molecular dynamics simulations were performed using the LAMMPS software[115]. The TTA-SPDFP and TTA-MCDFP COF structures were constructed from the CIF files, A 6x6 layer of 6 stacked sheets were placed in a vacuum slab of z = 9.5 nm. The simulation box along the x, y-axis adjusts such that PBC extends the simulation dimensions to higher length scales. The structures were then placed in a simulation box of dimensions 19.6 × 16.8 nm and solvated with water molecules, resulting in approximately 180,000 particles in each simulation box. The DREIDING force field[116] was used to represent the COFs, while the TIP3P model[117] was used to represent water. This parameter set has been previously shown to give good agreement with experimental data in our previous studies[118].

We investigated the behavior of the TTA-SPDFP and TTA-MCDFP COF structures in both vacuum and aqueous environments. For each COF system, the initial configuration was optimized by geometry optimization using the steepest descent for 5000 steps, allowing the system to reach its minimum energy. Then, we conducted MD simulations in an NPT ensemble, with the temperature set to 298.15 K and pressure set to 1 atm. Periodic boundary conditions were applied in all directions, allowing for volume and shape changes during the MD simulations. Pairwise interactions were truncated at a cutoff of 10 Å, and Ewald summation was employed for long-range Coulombic interactions with an accuracy of 1 × 10$^{-5}$ kJ/mol[119]. A time step of 2 fs was used, and simulations were run for approximately 25 ns for aqueous systems, with monitoring of convergence of the structural ensemble. The last 8 ns of the simulation were used for analysis. For vacuum simulations, the simulations were extended up to 100 ns, with the last 10 ns used for data analysis. The radial distribution functions were computed using the gmx_rdf module of GROMACS, and the bending angle was computed using the gmx_gangle module of GROMACS[120]. The membrane curvature was computed using the MDAnalysis Membrane Curvature

tool which calculates the Gaussian curvature averaged over the frames of the last 3 ns of the trajectory.

## Synthesis of dimethyl-4-bromopyridine-2,6-dicarboxylate (2a, Supplementary Fig. 1)

5.0 g of chelidamic acid hydrate was stirred with 25.0 g of $PBr_5$ and the reaction mixture heated at 90 °C for three hours. After the reaction completion, the reaction mixture was cooled to room temperature and diluted the reaction mixture with 60 mL of anhydrous chloroform. Then, the reaction mixture was cooled to 0 °C and 25 mL of methanol was added dropwise under stirring. After the addition of excess methanol to the reaction mixture, it was left to stir for an additional 30 min. The reaction mixture was filtered, washed with methanol and then dried. The product 2a obtained as a white solid (6.46 g, 86%). $^1$H NMR (500 MHz, $CDCl_3$, 25 °C): δ 4.02 (s, 6H, Ar-OC$H_3$), 8.45 (s, 2H, Ar-$H$); $^{13}$C NMR (500 MHz, $CDCl_3$, 25 °C): δ 53.4, 131.2, 135.1, 149.2, 164.1; MS (ESI-HRMS): m/z Calcd for $(C_9H_9BrNO_4)^+$: 273.9709 (M+H)$^+$, found: 273.9711 (M+H)$^+$.

## Synthesis of 4-bromo-2,6-pyridinedimethanol (2b, Supplementary Fig. 1)

6.39 g of dimethyl-4-bromopyridine-2,6-dicarboxylate in 125 mL absolute ethanol was stirred in the 250 ml round bottom flask and 4.25 g of $NaBH_4$ added in portions over a period of 15 min to the reaction mixture. Then, the reaction mixture was refluxed for 24 h, and allowed to cool to room temperature after the completion the reaction. The reaction mixture was extracted with ethyl acetate (3 x 30 mL). The organic phase was collected and dried over sodium sulphate and evaporated to dryness. The compound 2b was obtained in 62% (3.14 g) yield. $^1$H NMR (500 MHz, DMSO-d6, 25 °C): δ 4.51 (s, 4H, Ar-OC$H_2$), 7.52 (s, 2H, Ar-$H$); $^{13}$C NMR (500 MHz, $CDCl_3$, 25 °C): δ 64.1, 121.6, 133.9, 163.4; MS (ESI-HRMS): m/z Calcd for $(C_7H_9BrNO_2)^+$: 217.9811 (M+H)$^+$, found: 217.9812 (M+H)$^+$.

## Synthesis of 4-bromopyridine-2,6-dicarbaldehyde (2c, Supplementary Fig. 1)

1.43 g of 4-bromo-2,6-pyridinedimethanol was added 0.73 g of $SeO_2$ in 25 mL 1,4-dioxane under reflux condition for 20 h. Then, the reaction mixture was filtered through Celite and solvent was removed under reduced pressure. The product was purified by silica gel column using dichloromethane as an eluent. The product 2c was obtained as a pink solid in 69% yield (0.97 g) $^1$H NMR (500 MHz, $CDCl_3$, 25 °C): δ 8.31 (s, 2H, Ar-$H$), 10.14 (s, 2H, Ar-CH$O$); $^{13}$C NMR (500 MHz, $CDCl_3$, 25 °C): δ 128.8, 136.0, 154.0, 191.4; MS (ESI-HRMS): m/z Calcd for $(C_7H_5BrNO_2)^+$: 213.9498 (M+H)$^+$, found: 213.9499 (M+H)$^+$.

## Synthesis of 2-hydroxy-5-(4,4,5,5-tetramethyl-1,3,2-dioxaborolan-2-yl)benzaldehyde (4a, Supplementary Fig. 1)

Compound 4a was prepared according to the previously reported synthetic procedure with slight modification[121]. 5-bromo-2-hydroxybenzaldehyde (0.50 g, 2.5 mmol), bis(pinacolato)diboron (0.70 g, 2.75 mmol), potassium acetate (0.78 g, 8.0 mmol), and (1,1-bis(diphenylphosphino)ferrocene)-dichloropalladium(II) (0.055 g, 0.083 mmol) in 10 mL of 1,4-dioxane were added to a 50 mL microwave reaction vessel. The reaction mixture was refluxed under microwave irradiation for 15 min. After completion of the reaction, dioxane was removed and the residue was diluted with 20 mL of water and then extracted with dichloromethane (3 × 15 mL). The combined organic layers were dried under anhydrous magnesium sulphate, filtered, and then the solvent was removed under reduced pressure. The obtained crude product was purified using combi flash column chromatography with a hexane and ethyl acetate (10:1) gradient. Compound **4a** was obtained in 87% (0.53 g) yield. $^1$H-NMR (500 MHz, $CDCl_3$) δ 1.37 (s, 12H, -C$H_3$), 6.99 (d, 1H, $J$ = 8.4 Hz, Ar-$H$), 7.96 (dd, 1H, $J$ = 6.8 Hz, Ar-$H$), 8.07 (d, 1H, $J$ = 1.6 Hz, Ar-$H$), 9.94 (s, 1H, -C$H$=O),

11.24 (s, 1H, -O$H$); $^{13}$C NMR (125 MHz, $CDCl_3$, 25 °C): δ 24.9, 84.08, 117.1, 120.4, 141.4, 143.2, 164.0, 196.4; MS (ESI-HRMS): m/z Calcd for $(C_{13}H_{18}BO_4)^+$: 249.1292 (M+H)$^+$, found: 249.1289 (M+H)$^+$.

## Synthesis of 1′,3′,3′-trimethyl-6-(4,4,5,5-tetramethyl-1,3,2-dioxaborolan-2-yl)spiro[chromene-2,2′-indoline] (4b, Supplementary Fig. 1)

0.500 g (2.0 mmol) of compound 4a and 1,2,3,3-tetramethyl-3H-indol-1-ium iodide (0.606 g, 2.0 mmol) were dissolved in freshly distilled EtOH (10 mL) and anhydrous Et$_3$N (578 µL, 8.0 mmol). The reaction mixture was refluxed under argon for 3 h. After the completion of the reaction, the solvent was removed under reduced pressure. The obtained solid was purified using combi flash chromatography using hexane and ethyl acetate (1:19) gradient. Compound 4b was obtained in 71% (0.572 g) yield as pale pink solid. $^1$H-NMR (500 MHz, $CDCl_3$) δ 1.19 (s, 3H, -C$H_3$), 1.31 (s, 3H, -C$H_3$), 1.35 (s, 12H, -C$H_3$), 2.74 (s, 3H, -NC$H_3$), 5.67 (d, 1H, $J$ = 10.3 Hz, C=C-$H$), 6.53 (d, 1H, $J$ = 7.1 Hz, Ar-$H$), 6.72 (d, 1H, $J$ = 8.2 Hz, C=C-$H$), 6.84-6.91 (m, 2H, Ar-$H$), 7.08 (dd, 1H, $J$ = 6.4 Hz, Ar-$H$), 7.18-7.21 (m, 1H, Ar-$H$), 7.55 (d, 1H, $J$ = 1.5 Hz, Ar-$H$), 7.58 (dd, 1H, $J$ = 6.6 Hz, Ar-$H$); $^{13}$C NMR (125 MHz, $CDCl_3$, 25 °C): δ 20.0, 24.8, 25.9, 28.9, 51.7, 83.6, 104.6, 106.8, 114.6, 118.3, 119.0, 119.2, 121.5, 127.6, 129.5, 133.6, 136.8, 148.2, 157.3; MS (ESI-HRMS): m/z Calcd for $(C_{25}H_{31}BNO_3)^+$: 404.2391 (M+H)$^+$, found: 404.2389 (M+H)$^+$.

## Synthesis of (R)-4-(1′,3′,3′-trimethylspiro[chromene-2,2′-indolin]-6-yl)pyridine-2,6-dicarbaldehyde (SPDFP)

Compound 4b (0.75 g, 1.85 mmol), 4-bromo-2,6-diformylpyridine (2c) (0.396 g, 1.85 mmol) and Pd(PPh$_3$)$_4$ (0.286 g, 0.185 mmol) were added to a 100 mL three neck round-bottom flask under argon atmosphere. 20 mL of 1:1 THF/H$_2$O mixed solvent containing Na$_2$CO$_3$ (20 % solution) degassed solution was added to the reaction mixture. Then, the reaction mixture was refluxed at 85 °C for 20 h under an inert argon atmosphere. After completion of the reaction, 30 mL of distilled water was added to the reaction mixture and the aqueous solution was extracted with dichloromethane (3 × 25 mL). The collected organic layer was dried under anhydrous magnesium sulphate for 30 min. The resulting solution was filtered and concentrated to dryness. The resulting residue was purified by flash column chromatography [SiO$_2$: Hexane and CH$_2$Cl$_2$ (1:4)] to afford SPDFP as a light green color solid in 0.427 g (56%) yield. $^1$H NMR (500 MHz, $CDCl_3$, 25 °C): δ 1.22 (s, 3H, -C$H_3$), 1.35 (s, 3H, -C$H_3$), 2.79 (s, 3H, -NC$H_3$), 5.83 (d, 1H, $J$ = 10.3 Hz, C=C-$H$), 6.57 (d, 1H, $J$ = 7.8 Hz, Ar-$H$), 6.88-6.91 (m, 2H, Ar-$H$), 6.97 (d, 1H, $J$ = 10.2 Hz, C=C-$H$), 7.11 (dd, 1H, $J$ = 6.6 Hz, Ar-$H$), 7.21-7.24 (m, 1H, Ar-$H$), 7.52 (d, 1H, $J$ = 2.3 Hz, Ar-$H$), 7.55 (dd, 1H, $J$ = 6.1 Hz, Ar-$H$), 8.36 (s, 2H, Ar-$H$), 10.23 (s, 2H, -C$H$=O); $^{13}$C NMR (125 MHz, $CDCl_3$, 25 °C): δ 20.1, 25.9, 28.9, 52.0, 105.2, 106.9, 116.2, 119.4, 119.7, 120.9, 121.5, 121.9, 125.4, 127.5, 127.7, 128.6, 128.8, 136.4, 148.0, 150.4, 153.6, 156.7, 192.7; MS (ESI-HRMS): m/z Calcd for $(C_{26}H_{23}N_2O_3)^+$: 411.1704 (M+H)$^+$, found: 411.1707 (M+H)$^+$.

## Synthesis of 4,4′,4″-(1,3,5-triazine-2,4,6-triyl)-trianiline (TTA)

TTA was synthesized according to previously reported methods[122]. 1.54 g (13.08 mmol) of 4-aminobenzonitrile in a 50 mL oven-dry round bottom flask was stirred at 0 °C under an argon atmosphere and 4 mL of trifluoromethanesulfonic acid was added in dropwise for 10 min to the reaction mixture, maintaining the reaction temperature at 0 °C. Then, the reaction mixture was stirred at room temperature for 24 h in an argon atmosphere. After the reaction completion, 40 mL distilled water added to the reaction mixture and neutralized it by adding 2M NaOH solution until the pH reaches to 7. The obtained pale-yellow precipitate was filtered, wash several times with distilled water and then dried under vacuum. The pale-yellow solid of TTA obtained in 78 % yield. $^1$H NMR (500 MHz, DMSO-d$_6$, 25 °C): δ 5.91 (s, 6H, Ar-N$H_2$), 6.69 (d, 6H, $J$ = 8.7 Hz, Ar-$H$), 8.35 (d, 6H, $J$ = 8.7 Hz, Ar-$H$); $^{13}$C NMR (125 MHz,

DMSO-$d_6$, 25 °C): δ 113.6, 123.4, 130.6, 153.4, 170.1; MS (ESI-HRMS): m/z Calcd for $(C_{21}H_{19}N_6)^+$: 355.1665 (M+H)$^+$, found: 355.1667 (M+H)$^+$.

### Synthesis of 2,6-diformylpyridine (DFP)

DFP was synthesized by following the previously reported synthetic procedure[109]. 2,6-pyridinedimethanol (5.5 g, 39.5 mmol), and selenium dioxide (4.25 g, 38.0 mmol) in dioxane (100 mL) and water (4%) in a 250-mL round-bottom flask was refluxed for 20 h. After the completion of the reaction, the mixture was filtered in hot condition through celite bed. The filter was washed with hot dioxane (2 x 10 mL) and $CH_2Cl_2$ (2 x 10 mL). Solvents were removed from the filtrate under reduced pressure and the obtained solid was purified by silica gel column using $CH_2Cl_2$ as an eluent to afford DFP as a pale pink solid. Yield: 4.0 g, 76%. $^1$H NMR (500 MHz, CDCl$_3$, 25 °C): δ 8.09 (t, $J$ = 7.3 Hz, 1H, Ar-$H$), 8.17 (dd, $J$ = 7.3 Hz, 2H, Ar-$H$), 10.16 (s, 2H, Ar-C$H$O); $^{13}$C NMR (125 MHz, CDCl$_3$, 25 °C): δ 125.5, 138.6, 153.2, 192.6; MS (ESI-HRMS): m/z Calcd for $(C_7H_6NO_2)^+$: 136.0393 (M+H)$^+$, found: 136.0395 (M+H)$^+$.

### Synthesis of TTA-SPDFP COF

The spiropyran-functionalized COF (TTA-SPDFP) was prepared via imine condensation of the monomers 4,4′,4″-(1,3,5-triazine-2,4,6-triyl)-trianiline (TTA, 0.06 mmol, 21.24 mg), pyridine-2,6-dicarbaldehyde (DFP, 0.06 mmol, 8.1 mg), and (R)-4-(1′,3′,3′-trimethylspiro[chromene-2,2′-indolin]-6-yl)-pyridine-2,6-dicarbaldehyde (SPDFP, 0.03 mmol, 12.3 mg). The reactions were run in 1,4-dioxane using acetic acid (0.5 mL, 6 M) as a catalyst at room temperature under sonication in a 25 mL glass vessel at room temperature for 2 h. The resulting product was obtained as a yellow powder. The product was purified by washing with ethanol, followed by drying at 120 °C for 12 h. The product was completely insoluble in water and common organic solvents. Sonochemical reactions were performed with a 550 W ultrasonic probe in continuous mode using a 3 mm microtip probe at 40% sonication power (the maximum power recommended for the used probe).

## Data availability

Data discussed in this study are presented in the text and the Supplementary Information. Additional data can be obtained from the corresponding author upon request. Calculated structure of TTA-SPDFP and TTA-MCDFP COFs can be obtained from Figshare repository using the references https://doi.org/10.6084/m9.figshare.23451512 and https://doi.org/10.6084/m9.figshare.23451509.

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

## Acknowledgements

This work was supported by New York University Abu Dhabi and the NYUAD Water Research Center, funded by Tamkeen under the NYUAD Research Institute Award (project CG007). We thank NYUAD for their generous support for the research program. We thank Sandooq Al Watan for funding (Grant No. SWARD-S22-014, Project ID: PRJ-SWARD-628). The research work was carried out by using the Core Technology Platform resources at NYUAD. Computer simulations were carried out on the High-Performance Computing resources at New York University Abu Dhabi. S.K. and N.A. are funded by the NYUAD research fund AD181. N.S. acknowledge the funding sponsored by the Zayed Center for Health Sciences at the UAE University (Grant #12R113).

## Author contributions

G.D., T.P. and A.T. conceived and designed the experiments. G.D. and T.P. synthesized the starting materials. G.D. performed the COF synthesis and G.D., T.P. carried out all structural characterizations. S.K.S. and R.J. performed the AFM study of the COFs. R.P. and F.R. performed the HRTEM analysis. N.A. and S.K. performed the computational study. S.V. performed the solid-state NMR experiments and analyzed the data, B.G. helped to characterize the FT-IR analysis for the photochemical conversion, N.S. and F.C. performed the solid-state UV-vis and photoluminescence analysis. R.G.A. performed water adsorption measurements. M.A.O. was involved in the preparation of the manuscript and gave valuable suggestions. G.D., A.T. and M.A.O. wrote the manuscript, and all authors reviewed and commented on the manuscript.

## Competing interests

The authors declare no competing interests.
