## [Peer Review File · Nature Communications]

Light-Driven Self-assembly of Spiropyran-Functionalized Covalent Organic FrameworkReviewers' Comments:

Reviewer #1:

Remarks to the Author:

This is a borderline paper for Nature Communications. The photolight driven COF structure changes is of interest; however, no specific and unique properties has been developed other than the luminiscent changes. Furthermore, the PXRDs of the as-synthesized materials are of low quality, so the structure determination has some uncertainty.

Reviewer #2:

Remarks to the Author:

In this manuscript, the authors designed and synthesized spiropyran functionalized COF with reversible photoswitching behavior under the UV/Vis light irradiation. Interestingly, the COF shows a morphological transition in solution upon UV light irradiation. Study of the photoisomerization in COFs is of significance, however, more profound experiments must be done to publish this work.

1. The authors claimed "TTA-SPDFP is thermally stable up to 449 °C (Supplementary Fig. 5) without any significant loss in weight.", however, TG curve showed continuous weight loss below 449 °C.

Possible reasons should be given, and the authors could be more careful about the description.

2. The BET surface areas are quite low for these COF structures. What is the theoretical surface area for the COFs? How does photoisomerization of the spiropyran units in COF affect pore size? The authors should provide the pore size distributions of the COFs.

3. The PXRD patterns with broad peaks and low intensity, together with the low surface area, might suggest the poor crystallinity of the COFs. The authors could optimize the conditions to improve the quality of COFs.

4. The authors mentioned "The spiropyran moiety is very sensitive to moisture, and some of the spiro moieties can be converted to the merocyanine form in the presence of moisture in the absence of UV light." Since the morphological changes of the COFs in solution phase are investigated in aqueous dispersion, the factors that induce the changes can be complicated, and the influence of water should be studied.

5. The proposed mechanism for vesicle formation should be supported with characterizations and/or calculations, as this is an important part of the study of the self-assembly process.

6. The authors observed enhanced crystallinity of COF after irradiation as evidenced by HRTEM. How about the PXRD and gas sorption of the COF after irradiation?

7. In figure 4e, the authors present scheme showing the enhancement in water permeability of the COF after photoisomerization, however, there is no direct evidence for this point.

8. The potential applications of such changes in COFs could be further explored.

Reviewer #3:

Remarks to the Author:

In this work, a kind of spiropyran-functionalized COF (spiro-COF) was synthesized by using mixed linkers and sonochemical method. The resulting spiro-COF contained a spiropyran in each pore and displayed highly reversible photoswitching properties in the solid state upon UV/Vis light irradiation. The incorporation of spiropyran into COF led to photoregulated self-assembly in one step from rigid spheres to highly stable vesicles with enhanced luminescence properties and improved crystallinity upon light irradiation. The resulting spiro-COF displayed highly reversible acidochromic behavior with inappreciable structure change. Overall, this work is quite interesting and systematic, which will facilitate the development and application of emitting photoswitchable COF materials. The manuscript is well-organized in most part. Therefore, the manuscript may be acceptable after addressing the following issues.

1. The explanation for the novelty should be more convincing by identifying the major differences with

the previous work from the same research team.

2. "Without steric hindrance" is not a very strict description.

3. "weakly crystalline" can be revised to "low crystallinity"

4. "structural integrity" should be verified by application performance rather than UV-vis spectrum and SEM images.

We thank the reviewers of our manuscript for their constructive comments and for helping us to better understand our system. These comments were essential to better understand the mechanism of the photophysical properties of the material and to strengthen the manuscript. Below are our responses to the reviewers' comments.

Reviewers' comments:

Reviewer #1 (Remarks to the Author):

This is a borderline paper for Nature Communications.

Response: This is a borderline comment by a reviewer for Nature communications. We have considered the concerns expressed by the reviewer and have addressed those that have served to help strengthen the studies and results of this work.

1. The photolight driven COF structure changes is of interest; however, no specific and unique properties has been developed other than the luminescent changes.

Response:

In the revised manuscript, two different unique features of our photosensitive COF were developed, such as (i) dual functional light responsive self-erasing rewritable prints and (ii) its water harvesting applications.

In this work, we demonstrate TTA-SPDFP \leftrightarrow TTA-MCDFP COF's excellent switching reversibility, as shown in Figs. 8a and 8b, for temporary printing applications. The uniform dispersion of TTA-SPDFP COF was achieved by immersing the COF powder in ethanol under vigorous ultrasonic treatment for 20 minutes. Due to the high dispersibility of the COF material, it can be uniformly applied to the surface of a cellulose filter paper strip. The paper strip was air dried overnight at room temperature. Then, the coated paper was irradiated with 365 nm UV light through a honeybee design stencil mask for a few seconds. The stencil was then removed from the surface of the coated paper, revealing a clear, well-resolved, deep green print of the insect on the COF-coated paper (Fig. 8b), which was attributed to the formation of the merocyanine form (TTA-MCDFP COF). Due to the exceptional photostability of the TTA-MCDFP COF's metastable form, the printed sample can be kept for 2 days without fading of the color (Supplementary Fig. 23). Reversible isomerization of TTA-MCDFP \rightarrow TTA-SPDFP could also be achieved by thermal annealing of the COF-coated test strip. Thus, the printed content can be quickly erased by

heating the strip for 1 minute (60 °C) if needed, which could be used to print another designed mask (Figs. 8b and 8c). Therefore, our photochromic COF, which acts as an erasable ink, can be effectively used as a convenient and easy-to-use solution for temporary printing of tattoos, advertising stickers, and clothing designs.

We also investigated the water adsorption properties of the obtained COFs, as the hydrophobicity/hydrophilicity of TTA-SPDFP and TTA-MCDFP COFs affects the photoswitchable system. We performed water vapor adsorption analysis at room temperature for both the TTA-SPDFP and TTA-MCDFP COFs (Fig. 8d). The conversion of the highly hydrophobic form of TTA-SPDFP to the more hydrophilic form of TTA-MCDFP was illustrated by the change in the water adsorption isotherm from a typical type VII108 water adsorption isotherm with a total uptake of about 8%, as shown in Fig. 8d (black), to an S-shaped type V108 isotherm (Fig. 8d, red), showing enhanced sorbent-water interaction with a total uptake of about 20% at 97% RH. These results confirm that the indoline cation in merocyanine is more hydrophilic compared to the synthetic material, which promotes water uptake in the COF network. The water adsorption studies were repeated three times to verify the stability of the TTA-MCDFP COF (Supplementary Fig. 24). To our knowledge, light-mediated water adsorption in COFs is not yet known, and our photoactive TTA-MCDFP COF will open new avenues for the development of photo responsive water harvesters.

Fig. 8 Applications of TTA-SPDFP/MCDFP COFs a) Schematic representation of the reversible, light/heat-promoted isomerization of TTA-SPDFP COF. b) Photographs of Whatman 40 filter paper coated with TTA-SPDFP COF powder ($5 \times 5 \text{ cm}^2$) and the images were created sequentially in the same piece of paper by irradiating through different designed masks by alteration of light/heat (365 nm light for 5 seconds; $\Delta = 60 \text{ }^\circ\text{C}$). c) Reversible switching cycles in absorbance at ($\lambda_{\text{max}} = 664 \text{ nm}$) of TTA-SPDFP COF in response to UV light (365 nm, 5 seconds) and thermal heating ($60 \text{ }^\circ\text{C}$, 60 seconds). d) Water adsorption-desorption isotherms for the TTA-SPDFP and TTA-MCDFP COF powder as a function of relative humidity.

Supplementary Fig. 23. Test of fading stability of printed paper at different time intervals. The long fading time of the printed pattern indicates that TTA-MCDFP can be used for erasable inkless printing.

Supplementary Fig. 24. Recyclability and reusability of COFs for water adsorption. Water adsorption–desorption isotherms of TTA-MCDFP COF at three consecutive cycles, measured at 298 K.

2. Furthermore, the PXRDs of the as-synthesized materials are of low quality, so the structure determination has some uncertainty.

Response: The low intensity of the PXRD pattern of the TTA-SPDFP COF may be attributed to the presence of the bulky, non-planar spiropyran and methyl units within the pore channels, which are non-planar and thus prevent efficient π - π stacking. To confirm this hypothesis, an isostructural triazine-based COF, TTA-DFP, was synthesized using the

same synthetic protocol.⁸⁸ As shown by PXRD, the TTA-DFP COF has a higher crystallinity as compared to the TTA-SPDFP COF, likely due to the enhanced π - π stacking, as shown in Supplementary Fig. 6.

Supplementary Fig. 6. a) Comparison of the PXRD patterns of TTA-SPDFP and TTA-DFP. b) Simulated structures that resulted from an eclipsed AA stacking model for TTA-SPDFP and TTA-DFP. Weak PXRD intensity for TTA-SPDFP could be due to the distortion of the π - π stacking between the COF layers.

Fig. 1 b, c) Powder X-ray diffraction (PXRD) patterns of the TTA-SPDFP COF and TTA-MCDFP COF. The insets show the cartoon images of the spiropyran and merocyanine.

Reviewer #2 (Remarks to the Author):

In this manuscript, the authors designed and synthesized spiropyran functionalized COF with reversible photoswitching behavior under the UV/Vis light irradiation. Interestingly, the COF shows a morphological transition in solution upon UV light irradiation. Study of

the photoisomerization in COFs is of significance, however, more profound experiments must be done to publish this work.

Response: We thank reviewer # 2 for the constructive comments and suggestions that have helped to improve our work. We have now considered all of the suggested comments.

1. The authors claimed “TTA-SPDFP is thermally stable up to 449 °C (Supplementary Fig. 5) without any significant loss in weight.”, however, TG curve showed continuous weight loss below 449 °C. Possible reasons should be given, and the authors could be more careful about the description.

Response: We thank the reviewer for raising this point. For TGA, the weight loss below 449 °C is due to the presence of accessible atmospheric moisture trapped inside the COF network. The presence of atmospheric moisture in the COF network can be further confirmed by dynamic vapor sorption (DVS) analysis as a function of relative humidity, as shown in figure 8f. Notwithstanding the hydrophobic nature of the TTA-SPDFP COF, the material is still capable of adsorbing water (RH = 97% reaches 8.2 wt% at 298 K). The isotherm was found to be reversible and exhibits slight hysteresis, an indication that the COF material has some hygroscopic nature to it.

2. The BET surface areas are quite low for these COF structures. What is the theoretical surface area for the COFs? How does photoisomerization of the spiropyran units in COF affect pore size? The authors should provide the pore size distributions of the COFs.

Response: The permanent porosity of the TTA-SPDFP and TTA- MCDFP COFs was determined by measuring their N₂ adsorption isotherms at 77K, as shown in Supplementary Fig. 10 a and b. The surface area of the open form, TTA-MCDFP, displays a restricted adsorption behavior with a Brunauer-Emmett-Teller (BET) surface area value of 92 m²g⁻¹ compared to the closed form, TTA-SPDFP, with a surface area of 334 m²g⁻¹. As additional evidence for the conversion of SPDFP to MCDFP, non-local density functional theory (NLDFT) calculations show a decrease in total pore volume from 0.28 cm³g⁻¹ to 0.13 cm³g⁻¹, although no drastic changes in average pore sizes were observed, as shown in Supplementary Fig. 10 c and d. The decrease in the surface area in the merocyanine (MC) form is expected and due to the change in the size of the molecule as well as its orientation from orthogonal for SP to planar for MC. This transition causes the MC form to protrude more into the pores and partially block the pore channels, as shown in Supplementary Fig.

11 a and b. The theoretical surface areas of the TTA-SPDFP and TTA-MCDFP COFs were determined to be $2543 \text{ m}^2\text{g}^{-1}$ and $2474 \text{ m}^2\text{g}^{-1}$, respectively. These values are very high compared to the experimentally obtained surface areas, which is obvious since the simulated surface areas were measured for a purely crystalline system. However, the trend of decreasing BET surface area in the merocyanine form correlates with experimental observations. Thus, this statement was clearly justified in the main text. A similar decrease in N_2 adsorption upon light irradiation was also reported in spiropyran-based MOFs, *J. Mater. Chem. A*, **2016**, 4, 10816.

Supplementary Fig. 10. a, b) N_2 adsorption isotherms and c, d) pore size distribution of TTA-SPDFP and TTA-MCDFP COFs. Surface area measurements were performed by activating the samples at $85 \text{ }^\circ\text{C}$ for 24 hours under high vacuum.

Supplementary Fig. 11. Structural differences between a) TTA-SPDFP and b) TTA-MCDFP COFs. The yellow and green circles highlight the light-induced changes inside the pore.

3. The PXRD patterns with broad peaks and low intensity, together with the low surface area, might suggest the poor crystallinity of the COFs. The authors could optimize the conditions to improve the quality of COFs.

Response: We have now provided improved PXRD data as shown in Fig. 1b-c. The obtained PXRD patterns are similar to those of our previously published isostructural COFs, *Chem. Sci.*, **2018**, 9, 8382 and *J. Am. Chem. Soc.* **2019**, 141, 48, 19078–19087. As for the BET analysis, unfortunately, we could not obtain a higher surface area similar to other reported COFs. The low surface area of TTA-SP/MCDFP COF is expected because of the large bulky non-planar spiro group blocking the pore, and the weak crystallinity nature of the COF. These are the main reasons for the low surface area. A similar observation was also made in a recent report by Wang *et al.* in *Chem. Sci.* **2022**, 13, 5964. The low surface area in our photo-responsive COFs does not affect the explored unique properties of the COF material. The new PXRD and BET data are now included in the revised main text and supplementary information.

4. The authors mentioned “The spiropyran moiety is very sensitive to moisture, and some of the spiro moieties can be converted to the merocyanine form in the presence of moisture in the absence of UV light.” Since the morphological changes of the COFs in solution

phase are investigated in aqueous dispersion, the factors that induce the changes can be complicated, and the influence of water should be studied.

Response: We thank the reviewer for pointing out this critical observation. To investigate the influence of water on the morphological changes of the COF, we observed the morphological changes of the as-synthesized COF in both polar and nonpolar media (water and hexane). Interestingly, the TTA-SPDFP COF isomerizes to TTA-MCDFP COF in water under UV light. In water, the merocyanine isomer is extremely stable due to its highly polar nature. The polarity-induced photoisomerization in water is confirmed by the extremely strong luminescence properties, which are due to the ring-opened merocyanine form, and the luminescence can be switched from the open to the closed form upon thermal heating (Fig. 8d). The compound's detailed photophysical properties in water were discussed in response 1 to reviewer 1. To gain further insight into the morphological study, we also studied the morphological changes in the nonpolar medium hexane (supplementary Fig. 19). The TTA-SPDFP COF showed no morphological changes when dispersed in hexane. This confirms that water plays an important role in the self-assembly of spiropyran COF systems. Polarity-induced isomerization of spiropyran-based materials are well described in the literature, *Org. Lett.* **2009**, *11*, 1571–1574, *ACS Omega* **2021**, *6*, 35619, *ACS Phys. Chem Au*, **2023** doi.org/10.1021/acspyschemau.2c00067.

Supplementary Fig. 19. a) SEM image of the as-synthesized TTA-SPDFP COF. The SEM imaging sample was prepared by drop casting the diluted solution of TTA-SPDFP COF, dispersed in hexane, onto a Silicon wafer. b) The solution was then irradiated with UV light at 365 nm for 5 hours and SEM samples were prepared by drop casting. Microscopic analysis was performed at room temperature.

5. The proposed mechanism for vesicle formation should be supported with characterizations and/or calculations, as this is an important part of the study of the self-assembly process.

Response: The mechanism for the formation of vesicles was clarified by the time dependent microscopic investigation (Fig. 6) and detailed discussion was included in the main text.

To understand the molecular mechanism behind vesicle formation, we conducted a series of all-atom molecular dynamic simulations. We constructed six-layer COF structures using the TTA-SPDFP and TTA-MCDFP initial structures, with box dimensions extending to $19.6 \times 16.8 \times 9.5 \text{ nm}^3$. Figure 6a shows our explicit water simulation setup. The relatively large systems are aimed at observing macroscopic changes reported experimentally. We kept the pressure and temperature at 1 atm and 298 K, respectively, while allowing the simulation box dimensions to change. We monitored the structure of the COFs in a vacuum and in water. Details of our simulation setup are given in the supplementary materials. Figure 6b shows the initial and the equilibrium structure of TTA-SPDFP in a vacuum compared to TTA-MCDFP in water (Fig. 6c). Consistent with our experiments, we observed notable differences between the two COFs. Interestingly, TTA-SPDFP remained mostly planar with some structural undulations during simulations, while TTA-MCDFP in water showed notable bending (Fig. 6b versus 6c). To quantify the curvatures forming, we computed the average curvature from the trajectories. We observe higher curvature in TTA-MCDFP in water compared to TTA-SPDFP in a vacuum (Fig. 6d) further supports the experiments. Once we obtained good agreement between experiments and simulations, we examined the underlying mechanisms behind the formation of curvature in the case of TTA-MCDFP in water. To do so, we analyzed the changes in the solvent environment and the atomic structure of the COF layers. We contrasted TTA-SPDFP in a vacuum, TTA-SPDFP COF in water, and TTA-MCDFP COF in water to elucidate the roles of water and non-bonded interactions. Radial distribution functions computed between the atom types of the COFs and water showed that TTA-SPDFP is hydrophobic in nature (Supplementary Fig. 20). Interestingly, upon a change in the covalent structure (TTA-SPDFP to TTA-MCDFP COF), two sites became hydrophilic in nature (Supplementary Fig. 21 and Fig. 6e). The atoms in the linker (O_R and N_3) switched from non-polar to polar nature (Fig. 6e),

resulting in higher flexibility of the region. As a result of the increased polarity and opposite charge that the two atom types possess, we observe a shift in the stacked layers, such that to stabilize the layers the average distance between the two closest O_R-N_3 pairs change from 0.9 nm to 0.5 nm (Fig. 6f). We found that this inter-layer electrostatic bond formation activated by photo-excitation slides the layers, resulting in an overall change in interlayer stacking leading to curvature formation in TTA-MCDFP COF.

Fig. 6 All atom molecular simulation study of TTA-SPDFP and TTA-MCDFP in water and in vacuum. a) Simulation box of the molecular dynamics set up, b) TTA-SPDFP in vacuum initial structure and its structure at the end of the simulation (top and side views). c) Same as B this time for TTA-MCDFP in water. d) The structure of COFs is characterized by the average curvature plotted for the x-y plane. The color bars show the Gaussian curvature value for the two COFs. The larger the value the larger the curvature. e) Radial distribution function of O_R and N_3 atoms with water for both COFs in water. f) The geometrical change of TTA-MCDFP layers leading to curvature formation observed during MD simulations.

Supplementary Fig. 20. Radial Distribution Function, $g_{\alpha\beta}(r)$, of TTA-SPDFP between atom type (α) with water atoms (β). The distance correlations of TTA-SPDFP atoms and water oxygen atom is shown in blue color, and TTA-SPDFP atoms and hydrogen of water is shown in red.

Supplementary Fig. 21. Radial Distribution Function, $g_{\alpha\beta}(r)$, of TTA-MCDFP between atom type (α) with water atoms (β). The distance correlations of TTA-MCDFP atoms and water oxygen atom is shown in blue color, and TTA-MCDFP atoms and hydrogen of water is shown in red.

6. The authors observed enhanced crystallinity of COF after irradiation as evidenced by HRTEM. How about the PXRD and gas sorption of the COF after irradiation?

Response: Both the PXRD and BET data of the light irradiated TTA-SPDFP COF are now included in Fig. 1 b, c and Supplementary Fig. 9 in the revised manuscript and Supplementary Information. After light irradiation of the solid samples, the crystallinity of the material did not change appreciably. However, the BET surface area of TTA-SPDFP COF is drastically reduced, and the light-irradiated PXRD and BET studies of TTA-SPDFP COF were discussed in response 2 to reviewer 1 and response 2 to reviewer 2.

7. In figure 4e, the authors present scheme showing the enhancement in water permeability of the COF after photoisomerization, however, there is no direct evidence for this point.

Response: To investigate the water permeability, we have included the water adsorption analysis before and after light irradiation in the solid state (Fig. 7f) in the revised manuscript.

8. The potential applications of such changes in COFs could be further explored.

Response: We have now provided two potential applications, such as inkless printing and light-controlled water harvesting with our COFs. More detailed results were discussed in response 1 to reviewer 1.

Reviewer #3 (Remarks to the Author):

In this work, a kind of spiropyran-functionalized COF (spiro-COF) was synthesized by using mixed linkers and sonochemical method. The resulting spiro-COF contained a spiropyran in each pore and displayed highly reversible photoswitching properties in the solid state upon UV/Vis light irradiation. The incorporation of spiropyran into COF led to photoregulated self-assembly in one step from rigid spheres to highly stable vesicles with enhanced luminescence properties and improved crystallinity upon light irradiation. The resulting spiro-COF displayed highly reversible acidochromic behavior with inappreciable structure change. Overall, this work is quite interesting and systematic, which will facilitate the development and application of emitting photoswitchable COF materials. The manuscript is well-organized in most part. Therefore, the manuscript may be acceptable after addressing the following issues.

Response: We thank the reviewer for their very positive comments and for helping us to improve the quality of the manuscript. We have addressed all of the reviewer's comments.

1. The explanation for the novelty should be more convincing by identifying the major differences with the previous work from the same research team.

Response: In our previous report on light responsive azo-functionalized COFs (J. Am. Chem. Soc. 2019, 141, 48, 19078) we mainly studied the light-responsive properties in the solution phase and did not investigate visual photoswitching/applications in the solid state, which is very important for practical applications. Reversible light switching in the solid state in a crystalline system is very challenging due to the crowded ordered structure. In this work, we provide a solution to the crowding effect that inhibits the photoisomerization process while maintaining the photoisomerization property of the SP linker without photodegradation. We adopted the mixed monomeric linker approach to prepare the reactive COF and successfully synthesized a spiropyran-functionalized COF (TTA-SPDFDP COF), which exhibited reversible photoswitching behavior in the solid state. We were able to induce the formation of three stable forms of spiropyran isomers by various external stimuli. Since we were able to isolate a photosensitive COF that exhibits visible photoswitching behavior in the solid state and exceptional photostability of the merocyanine form in the TTA-MCDFP COF, we have succeeded in developing a COF that meets the requirements for an inkless printing application. TTA-MCDFP COF in the solid/solution state also offers other potential applications, such as temperature sensing and water harvesting applications. Moreover, in our previous work on azo-functionalized COF, we did not observe any morphological changes between the two forms; however, in our present work, we obtained an unprecedented vesicular morphology with enhanced luminescent properties in water. The photostability of the TTA-MCDFP COF in water with vesicular morphology could be very useful for biomedical applications in the near future. All these properties make our stimuli responsive COFs even more interesting and open a new avenue for numerous applications.

We have added some statements to the introduction to highlight some of the advantages of the new material.

2. “Without steric hindrance” is not a very strict description.

Response: We have replaced the statement “Without steric hindrance” with “without any restriction”

3. “weakly crystalline” can be revised to “low crystallinity”

Response: We have changed the term “weakly crystalline” to “low crystallinity” as suggested by the reviewer.

4. “structural integrity” should be verified by application performance rather than UV-vis spectrum and SEM images.

Response: We have now provided three unique applications using our COFs to demonstrate their stability. Interestingly, our COFs showed excellent performance with no loss of structural integrity over multiple cycles, for details please read response 1, comment 1 for reviewer 1.

Reviewers' Comments:

Reviewer #1:

Remarks to the Author:

The authors have made quite a lot efforts to developing new functions and properties. This reviewer is happy with their attitudes and results. It can be accepted now.

Reviewer #2:

Remarks to the Author:

In the revised manuscript, although the PXRD data of COF was improved in the revised manuscript, the as-synthesized TTA-SPDFP-COF still exhibits limited crystallinity and low porosity, implying a high degree of amorphousness in its structure. In addition, the authors further showed potential applications of the COF in inkless printing media and water harvesting. However, the limited solubility and processability of COF make it challenging and less practical to use as ink compared to reported spiropyran-functionalized polymers. While the photomediated polarity change resulting from this approach has potential for water harvesting applications, the low water absorption capacity of the COF limits its effectiveness in this field. They do not prove the uniqueness and importance of the TTA-SPDFP-COF in the exhibited applications. Therefore, I do not recommend publishing this in Nature Communications.

Reviewer #3:

Remarks to the Author:

The revisions address my major concerns quite well and I have no further comments now.

Reviewer #1 (Remarks to the Author):

The authors have made quite a lot efforts to developing new functions and properties. This reviewer is happy with their attitudes and results. It can be accepted now.

Response: We sincerely thank Reviewer # 1 for their recommendation to publish. We agree that in addressing the reviewers' comments the level of the manuscript has been raised substantially.

Reviewer #2 (Remarks to the Author):

In the revised manuscript, although the PXRD data of COF was improved in the revised manuscript, the as-synthesized TTA-SPDFP-COF still exhibits limited crystallinity and low porosity, implying a high degree of amorphousness in its structure. In addition, the authors further showed potential applications of the COF in inkless printing media and water harvesting. However, the limited solubility and processability of COF make it challenging and less practical to use as ink compared to reported spiropyran-functionalized polymers. While the photomediated polarity change resulting from this approach has potential for water harvesting applications, the low water absorption capacity of the COF limits its effectiveness in this field. They do not prove the uniqueness and importance of the TTA-SPDFP-COF in the exhibited applications. Therefore, I do not recommend publishing this in Nature Communications.

Response: We thank Reviewer #2 for their time and efforts in reviewing our revised manuscript. Regarding the crystallinity of the COF, we tried several synthetic protocols to enhance the crystallinity of the material and the PXRD pattern of TTA-SPDFP-COF presented in our revised manuscript is the best that was achieved. We do agree with the reviewer that the crystallinity of the material is not as high as observed for other COFs; however, this is due to the structural geometry of the monomer that was used. The bent-shaped geometry of the aldehyde used to synthesize TTA-SPDFP-COF introduces geometric restrictions on the layers that are formed, resulting in the formation of less crystalline materials as compared to previously reported linear linkers that give rise to DFP-based quasi-honeycomb layered COFs having a similar structural topology.¹⁻⁴ As can be seen from its PXRD pattern, the TTA-SPDFP-COF demonstrated an obvious weak diffraction at $2\theta = 5.8^\circ$, correlating to a reflection from the (110) plane. This observation indicates the existence of some long-range molecular order which cannot be completely ignored. In addition, the presence of the dangling structurally non-planar spiro-groups located within the pore channels of the COF network causes distortions throughout the COF network, resulting in a less crystalline framework as mentioned in the manuscript. We also observe, as expected, the effect of the spiro-groups directed into the pore on the low surface area of the TTA-SPDFP COF. Similarly, low BET surface areas were also noted in our previous reports^{2,5} on azo/thio-anisole-based COFs.⁶ Very recently Wang et al. also reported a similar photo-responsive COF substituted with a bulky photo-responsive moiety directed inside

the pore and this material also had a low BET surface area of $54 \text{ m}^2 \text{ g}^{-1}$.⁶ The presence of bulkier spiro substituents unavoidably blocks the COF's pore channels resulting in a decrease in the surface area⁷ and partially weakening the crystallinity of COF's frameworks.⁸ As such, regarding the crystallinity and porosity of the COF, our recorded observations are simply part and parcel of this high functional class of stimuli-responsive material.

In terms of printing media applications, we would like to clarify that our study involves coating strips of paper strip with dispersed TTA-SPDFP COF and exposing it to UV-light as a substitute for ink, a process which is amenable to being widely accessible worldwide. Furthermore, conventional polymeric or small molecule materials used in printing applications for the production of inkless and erasable prints suffer from a lack of long-term stability. These materials tend to revert back to their original color within a few minutes after being exposed to light. However, our photochromic COF has many advantages including i) long-term photo-stability of the merocyanine form (TTA-MCDFP COF) within the ordered network, as indicated by the retention of the photogenerated color for 1 to 3 days; (ii) the reversibility of the photo-triggered color change on the same paper can be used for multiple cycles; and (iii) the ease of coating as the TTA-SP DFP COF is highly dispersible in ethanol, hence in response to UV-light (365 nm) the printed content is visible with high clarity. Given all of these notable advantages mentioned, we strongly believe that our TTA-SP DFP COF has the potential to revolutionize printing technology applications in the future.

As for the polarity-induced water harvesting applications, yes, we agree with Reviewer #2 that the water adsorption of TTA-MCDFP COF does not reflect high uptake, but it is still comparable to other related COF materials.⁶ On other hand, we would like to highlight that the water adsorption experiments were mainly conducted to understand the role of water in the self-assembly mechanism and the role of the MC in enhancing the hydrophilicity of the material as shown in the manuscript.

Overall, we thank Reviewer #2 for all the comments and we hope that we have managed to address his/her concerns.

Reviewer #3 (Remarks to the Author):

The revisions address my major concerns quite well and I have no further comments now.

Response: We deeply thank reviewer # 3 for their supportive remarks on our manuscript. The impact of the manuscript has certainly been elevated.

- 1 Das, G. *et al.* Covalent organic nanosheets for bioimaging. *Chemical Science* **9**, 8382-8387, doi:10.1039/C8SC02842G (2018).
- 2 Das, G. *et al.* Azobenzene-Equipped Covalent Organic Framework: Light-Operated Reservoir. *Journal of the American Chemical Society* **141**, 19078-19087, doi:10.1021/jacs.9b09643 (2019).
- 3 Benyettou, F. *et al.* Covalent Organic Framework Embedded with Magnetic Nanoparticles for MRI and Chemo-Thermotherapy. *Journal of the American Chemical Society* **142**, 18782-18794, doi:10.1021/jacs.0c05381 (2020).
- 4 Krishnaraj, C. *et al.* Triggering White-Light Emission in a 2D Imine Covalent Organic Framework Through Lanthanide Augmentation. *ACS Applied Materials & Interfaces* **11**, 27343-27352, doi:10.1021/acscami.9b07779 (2019).
- 5 Das, G. *et al.* Hydrophobicity Tuning in Isostructural Urchin-Shaped Covalent Organic Framework Nanoparticles by Pore Surface Engineering for Oil–Water Separation. *ACS Applied Nano Materials* **5**, 13745-13751, doi:10.1021/acsanm.2c00746 (2022).
- 6 Chen, Y. *et al.* A visible light/heat responsive covalent organic framework for highly efficient and switchable proton conductivity. *Chemical Science* **13**, 5964-5972, doi:10.1039/D2SC02100E (2022).
- 7 Ye, W. *et al.* Halide Perovskite glues activate two-dimensional covalent organic framework crystallites for selective NO₂ sensing. *Nature Communications* **14**, 2133, doi:10.1038/s41467-023-37296-0 (2023).
- 8 Wang, Z. *et al.* PolyCOFs: A New Class of Freestanding Responsive Covalent Organic Framework Membranes with High Mechanical Performance. *ACS Central Science* **5**, 1352-1359, doi:10.1021/acscentsci.9b00212 (2019).